

**Assessment of Glacier Area Change in the Tekes River Basin, Central Tien Shan,**
**Kazakhstan Between 1976 and 2013 Using Landsat and KH-9 Imagery**
Zamira Usmanova [1,2,3], Maria Shahgedanova [3], Igor Severskiy [1], Gennady Nosenko [4], Vassiliy
Kapitsa [1]
[1] Institute of Geography, Almaty, 050010, Kazakhstan;
[2] al-Farabi Kazakh National University, Almaty, 050040, Kazakhstan;
[3] Department of Geography and Environmental Science and Walker Institute for Climate System
Research, University of Reading, Whiteknights, Reading RG6 6AB, UK;
[4] Laboratory of Glaciology, Institute of Geography of the Russian Academy of Sciences, 29
Staromonetny Pereulok, Moscow, 119017, Russia.
*Correspondence to:* Z. Usmanova (zamira_usmanova@mail.ru)
**Abstract**
Changes in glacierized area in the Kazakhstani sector of the Tekes River basin were assessed
using Landsat and KH-9 imagery from 2013, 1992 and 1976. Between 1992 and 2013, the
combined area of 118 glaciers declined from $121.4 \pm 9.2$ km$^2$ to $105.0 \pm 5.5$ km$^2$. The total area
loss was $16.4 \pm 5.9$ km$^2$ or $13.5 \pm 7.5\%$. The rate of area reduction was 0.78 km$^2$ a$^{-1}$ or 0.64% a$^{-1}$.
This rate is lower than in other regions of northern Tien Shan because of the presence of several
large glaciers in the sample. The combined glacier area in 2013 exceeds the combined glacier
area reported by the RGI5.0 / GAMDAM inventories for 1999-2003 by 24% because the latter
did not include glacierized areas on slopes exceeding 40$^o$ and a number of small glaciers.
Changes in the recession rates between 1976, 1992 and 2013 were examined using a sub-sample
of 28 glaciers which occupied 61% of the total glacierized area in 1992 and 64 % in 2013. These
glaciers lost $8.3 \pm 5.6\%$ in the 1976-1992 period, $8.4 \pm 5.9\%$ in the 1992-2013 period and
$16.0 \pm 5.8\%$ between 1976 and 2013. The recession rates were $0.52 \pm 0.35\%$ a$^{-1}$ in 1976-1992 and





0.40±0.28% a$^{-1}$ in 1992-2013 and although they appear to indicate a slow down in the glacier
recession, the change in the retreat rates is within the uncertainty of measurement. The relative
reduction in glacier area in the sub-sample is lower than for the basin as a whole because of a
larger size of glaciers. Temperature increase was observed in all seasons reaching 0.18ºC per 10
years in summer and 0.39ºC per 10 years in autumn in the 1947-2015 period. Precipitation
exhibited strong variability declining between 1952 and 1977 and then increasing until 2000s
with a number of dry years in the 2010s. There was no statistically significant difference
between the means of annual precipitation in the 1952-1977 and 1977-2015 periods. Combined
with the nearly steady recession rates, this suggests that it is an increase in summer, late spring
and early autumn temperature that drives glacier retreat.
**1 Introduction**
The Tien Shan mountains are one of the main centres of contemporary glaciation in Eurasia
where at present glaciers occupy, according to different estimates, between 15,416 km$^2$ to 16,427
km$^2$ (Sorg et al., 2012). Glacier shrinkage has been observed in the region since the end of the
Little Ice Age (Solomina et al., 2004; Kutuzov and Shahgedanova, 2009) and recent assessments
suggest that overall, the Tien Shan lost 18±6% and 27±15% of glacierized area and mass
respectively between 1961 and 2012 (Farinotti et al., 2015). The rates of glacier recession vary
temporally and spatially because of complex topography, regional climatic differences and
variability in characteristics of glaciers potentially leading to variability in impacts of glacier
retreat such as changes in runoff and formation of glacier lakes. Table A1 presents glacier
recession rates as documented in the published literature illustrating both geographical and
temporal differences. The high retreat rates were observed in the southern Djungarskiy Alatau
and in Zailiyskiy Alatau (Solomina et al., 2004; Severskiy et al., 2016). At higher elevations in
the inner regions of the Tien Shan, glacier recession was slower. In the Saryjaz Ridge, the values
of glacier area change between 1990 and 2010 were close to the accuracy of measurements



(Osmonov et al., 2013). Pieczonka and Bolch (2015) reported similarly low recession rates for
the Kokshal-Too, Tomur and Inylchek regions and the Aksu catchment for the 1975-2008 period
as did Shannugan et al. (2009) for the Tarim basin in the Chinese Tien Shan by for the 1960s –
2000 period.
Most studies assess changes in the glacier extent for a single period (Table 1A). However, a
number of studies examining changes in recession rates over time highlight the acceleration of
glacier retreat in the last three decades, e.g. Aizen et al. (2006; 2007) in the Ala-Archa region
and Kutuzov and Shahgedanova (2009) in the Terskey-Alatoo. Having analysed Landsat, Corona
KH-4B and Hexagon KH-9 imagery, Narama et. al. (2010) reported a small acceleration of
glacier retreat in the western (Pskem) and northern (Ili – Kungey) Tien-Shan in the 2000-2007
period in comparison with the 1970-2000 period although it is not clear whether the reported
increase in the recession rates exceeds the uncertainty of measurements. By contrast, Severskiy
et al. (2016) showed that in the northern Tien Shan between 1955/56 and 1975 glacier recession
rates were comparable with the 1990-2008 period and in the 1970s, they were 2-3 times higher.
The earlier data in this study, however, were derived from the Catalogue of Glaciers of the
USSR based on the aerial photography and topographic maps which were not preserved and,
therefore, assessment of uncertainty in the earlier data was problematic.
The review of the existing studies (Table 1A) shows that although there were many assessments
of glacier change in the Tien Shan, given its strong spatial variability, it is important to generate
up-to-date detailed regional assessments of glacier change over different time periods for the
regions which were not examined so far using materials which enable uncertainty assessment.
One of the glacierized regions of the Tien Shan where glacier change has not been documented
in detail is the basin of the River Tekes in the Central Tien Shan within the national borders of
Kazakhstan (Fig. 1). The Tekes is a transboundary river originating in Kazakhstan, crossing into
China where, in confluence with the River Kash, it forms the River Ile, which, in turn, returns to
Kazakhstan as a major source of water for irrigation and the nourishment of Lake Balkhash. The





Tekes is nourished by snow and glacier runoff and, given the current requirements for water in
both countries, it is important to assess changes in the extent of glaciers in its basin.
The history of glacier research in the Tekes basin dates back over a century when Merzbacher in
1904 and 1905 described two large glaciers in the upper reaches of the Bayankol River on the
northern slopes of the Katta-Ashutor and the Saryjaz which were catalogued as Mramornaya
Stena (No 94), Simonov (No 89) and Bayankol (No 91) glaciers in the Catalogue of Glaciers of
the USSR (Vilesov, 1969). The first larger scale assessment was conducted in 1915,
documenting 74 glaciers in the Kazakhstani sector of the Tekes basin with the combined area of
116 km$^2$ and this was followed by a comprehensive inventory of 1956 which provided data for
the Catalogue of Glaciers of the USSR (Vilesov; 1969). More recently, Vilesov (2006) reported
15.8 % (0.45% a$^{-1}$) glacier area reduction between 1956 and 1990. Xu et al. (2015) reported a
reduction of 18% (0.37 % a$^{-1}$) in the Chinese sector of the Tekes basin between 1960 and 2009,
which is higher than in the neighbouring Saryjaz region (Osmonov et al., 2013). There are no
post-1990 assessments of glacier change in the Kazakhstani sector of the Tekes basin.
Glacier outlines from the 1999-2003 period are available from the Randolph Glacier Inventory
5.0 (RGI5.0; http://www.glims.org/RGI/rgi50_dl.html). These outlines were generated by the
Glacier Area Mapping for Discharge from the Asian Mountains (GAMDAM) project using
manual mapping of glaciers on the Landsat imagery (Nuimura et al., 2015). Although the Tekes
basin was included in the GAMDAM inventory, the main purpose of GAMDAM was to derive
the extent of glacierized area for the High Asia and its much larger geographical units rather and,
therefore, no analysis for the Tekes basin is presented.
The objectives of this paper are: (i) to present inventories of glaciers derived from the Landsat
imagery for the Kazakhstani sector of the Tekes basin for 1992 and 2013; (ii)) using a sub-
sample of glaciers, analyse changes in their extent between 1976, 1992 and 2013 and compare
the retreat rates; (iii) analyse changes in the extent of glacierized area and discuss them in the
context of climatic change and variability.




## 2 Study area


In the Kazakhstani sector of the Tekes basin, glaciers are located between 42º43´N and 40º16´N
and 79º13´E and 80º20´E on the northern macroslopes of the Terskey Alatoo and Saryjaz Ridges,
on the western macroslope of the Meridional Ridge and in the Katta Ashutor Ridge (Fig. 1).
Elevations in the Terskey Alatoo are mostly within 4200-4400 m a.s.l. range increasing to 5000-
5200 m in the Katta Ashutor and 5700-6100 m in the Saryjaz. In the Meridional Ridge,
elevations increase from 4000 m at the source of the Narynkol River to about 6000 m in the
south (Fig. 1).

According to the Catalogue of Glaciers of the USSR (Vilesov, 1969), in 1956 there were 152
glaciers with a combined area of 143.0 km$^2$ in the study area. The largest glaciers were located in
the Katta-Ashutor and the Saryjaz (e.g. Simonov, Bayankol, Mramornaya Stena with individual
areas of 28.1 km$^2$, 6.9 km$^2$, and 22.5 km$^2$ respectively) and on the northern macroslope of the
Meridional Ridge (e.g. Sauruksaiskiy (No 104), 7.9 km$^2$). In all, in 1956, there were three
compound valley glaciers with a combined area of 58.5 km$^2$ and position of glacier tongues at
approximately 3350 m, 21 valley glaciers with a combined area of 54.4 km$^2$ and 19 cirque-valley
glaciers with a combined area of 14.0 km$^2$. The valley glaciers descended to approximately 3550
m. There were 27 cirque and 38 hanging glaciers with the combined areas of 6.9 km$^2$ and 5.6
km$^2$ and two ice aprons (1.8 km$^2$) and one flat-summit (0.2 km$^2$) glaciers.

The climate of the area is characterised by strong seasonal variations in atmospheric circulation
dominated by the western extension of the Siberian anticyclone in winter whose influence is
stronger in the valleys and diminishes with altitude giving way to the westerly flow
(Panagiotoupulos et al., 2005). The thermal Asiatic depression dominates in summer and the
westerly flow in spring and autumn with frequent depressions in September-October. These
changes predetermine strong seasonal fluctuations in temperature and precipitation (Fig. 2).





In winter, the combined effect of the Siberian anticyclone and high elevations results in low
temperatures whose December-January-February (DJF) means range between -10.8ºC at 1800 m
at Narynkol meteorological station (Fig. 2; Table 1) and about -20ºC at 3600-4000 m. The June-
July-August (JJA) mean temperature is 15.2ºC at Narynkol decreasing to 2-4ºC at the glacier
tongue elevation where positive air temperatures are observed between early June and early
September. The ablation season is normally limited to JJA. Annual precipitation increases from
about 390 mm at Narynkol to 1000-1200 mm at the glacier elevation of 3600-4000 m (Vilesov,
1969). Precipitation maximum is observed in late spring - summer while winter precipitation is
low (Fig. 2; Table 1). At Narynkol, the May-August precipitation was 217 mm accounting for
56% of the annual total while DJF precipitation is 35 mm accounting for 9%. Accumulation
occurs throughout the year.

**3 Data and methods**
**3.1 Satellite imagery and glacier mapping**
Changes in glacierized area were assessed for the Kazakhstani sector of the Tekes River basin
(Fig. 1) using Landsat imagery from 1992 and 2013 (Table 2). Earlier KH-9 Hexagon imagery
was available for January 1976 and used for a limited number of glaciers whose tongues were
free of snow cover due to the strong negative precipitation anomalies in 1975 which was the
driest year on record with annual precipitation two standard deviations below the record mean
(see Fig. 9). Scenes from the same acquisition were used by Pieczonka and Bolch (2015) for the
calculation of geodetic mass balance using satellite imagery in the Tien Shan.
For 1992 and 2013, areas of 118 glaciers were mapped of which the largest was 21.3 km$^2$ and the
smallest was 0.01 km$^2$. Four Landsat scenes (Table 2) were obtained from the US Geological
Survey (USGS; http://glovis.usgs.gov/) in the Universal Transverse Mercator (UTM) zones 44
WGS 84 projection. All Landsat images were acquired under [nearly] cloud-free conditions. The
2013 image was acquired at the end of the ablation season. The 1992 image was obtained at the



middle of the ablation season, however, at the time of image acquisition glacier tongues were
free of seasonal snow and the image was suitable for glacier mapping.
Glacier outlines were mapped using Landsat bands 7, 5, 3 for 1992 and 2013 and 8
(panchromatic) for 2013. Manual on-screen mapping was used despite the advantages of
automated mapping demonstrated by Paul et al. (2009; 2013). This is because relative error
strongly increases with decreasing glacier area (Paul et al., 2013; Fischer et al., 2014) and with
the presence of debris cover (Bhambri et al., 2011; Bolch et al., 2008; Racoviteanu et al., 2008;
Frey et al. 2012; Paul et al., 2013). Paul et al. (2013) and Fischer et al. (2014) have shown that
the bias significantly increases for glaciers with areas less than 1 km$^2$, which constitute 21% of
all glaciers in the Tekes basin, reducing the advantages of automated techniques. For the same
reasons, manual mapping of glacier boundaries was used in the GAMDAM inventory (Nuimura
et al., 2015) and by Narama et al. (2010) for the inventory focusing on the western, central and
northern Tien Shan. To assist manual delineation of debris-covered snouts, the higher-resolution
imagery from Google Earth was inspected in conjunction with the Landsat images.
Most glaciers in the study area have clearly defined ice divides but, where required, ASTER
DEM    obtained    from    ASTER    Global    Digital    Elevation    Model    site
(http://gdem.ersdac.jspacesystems.or.jp/) was used to delineate the upper boundaries. This
delineation was consistent with that used in the Catalogue of Glaciers of the USSR (Vilesov,
1969) enabling a comparison of glacier change since 1956. It was assumed that the upper
boundaries of the glaciers did not change between 1992 and 2013. The areas of emerging rocks
in the upper sections of glaciers were mapped and their areas were deduced from the glacier area.
For glaciers that fragmented between 1992 and 2013, combined areas were recorded. There were
no known surging glaciers in the study region.
Tongues of 28 glaciers were clearly visible on KH9-Hexagon image (Table 2) allowing mapping
of their positions. Technical details of KH-9 Hexagon imagery, declassified in 2002, are
provided by Surazakov and Aizen (2010) and Burnett (2012). The sensor used a frame mapping



camera with a 23 x 46 cm frame and a focal length of 30.5 cm. The KH9 scenes covered areas of
125 x 250 $km^2$ on a scale of 1:600,000 at an altitude of ~170 km (Burnett, 2012). The images
were provided by the USGS with a scan resolution of about 14 μm. The pre-processing of the
KH-9 image, involving the removal of internal film distortions based on reseau crosses, has been
done following Pieczonka et al. (2013). The KH-9 image were co-registered to the orthorectified
Landsat TM and Landsat OLI TIRS images using a network of GCPs that have been collected
from the Landsat images. This procedure was carried out using ERDAS Imagine 9.0 software
and produced the maximum root-mean-square error ($RMSE_{xy}$) values of 5.1 m and 4.8 m
respectively. Following the co-registration, ice margins in the glacier ablation zones were
manually derived from the KH-9 images. Glacier margins in the accumulation zone were
delineated from the Landsat imagery. Figure 3 shows an example of a comparison of glacier
outlines from the Landsat and KH-9 images.

**3.2 Quantification of uncertainty**
**3.2.1 Landsat images**
For each scene, the accuracy of the orthorectification was verified using a network of interactive
ground control points (GCPs) obtained from 1:50000 maps using clearly identifiable terrain
features whose location did not change. For each glacier, two uncertainty terms have been
calculated resulting from the uncertainty of orthorectification and from identification of the
glacier margins. The uncertainty of orthorectification was calculated following Granshaw and
Fountain (2006). A buffer, with a width of half of the $RMSE_{x,y}$ was created along the glacier
outlines and the uncertainty term was calculated as an average ratio of the original glacier areas
to the areas with a buffer increment. The values of $RMSE_{x,y}$ were 25 m and 29 m for the Landsat
TM scenes and 14.5 m and 14 m for Landsat OLI TIRS scenes resulting in a mean uncertainty of
±18.1% and ±12.3% for 1992 and 2013 respectively. The uncertainty of glacier margin
identification was taken as 3.5% for each image following a multiple digitization study by Paul





et al. (2013). The total mean uncertainties of glacier map area calculation were ±18.6% and
±13% in 1992 and 2013 respectively.
To estimate uncertainty of glacier area change, the 1992 and 2013 scenes were co-registered
using 15-20 well-identifiable points on the images. The maximum value of $RMSE_{x,y}$ was 5 m.
The uncertainty of co-registration was calculated using the buffer method resulting in an average
uncertainty of ±3.1% and ±4.1% for 1992 and 2013 respectively. The combined uncertainty of
co-registration and ±3.5% uncertainty of glacier margin identification was ±7.5%.
Debris cover on glacier snouts is a major source of uncertainty in glacier mapping (Bolch et al.,
2008; Racoviteanu et al., 2008; Frey et al. 2012; Paul et al., 2013). We considered and rejected
the frequently used practice of inflating the uncertainty term of glacier margin identification for
the debris-covered sectors of glacier tongues (e.g. Frey et al., 2012; Shahgedanova et al., 2014).
This is because debris cover is extensive on the tongues of the largest glaciers (e.g. Bayankol,
Mramornaya Stena, Simonov; Fig. 7 further in the text) where it does not merge periglacial
landforms and a close inspection and the use of higher resolution Google Earth imagery enables
the delineation of glacier margins. On other glaciers, the extent of debris cover is significantly
smaller than in the neighbouring Saryjaz region (Osmonov et al., 2013) enabling its manual
delineation.

**3.2.2 KH-9 images**
To estimate the uncertainties of mapping of glacier area from the 1976 KH-9 imagery, we
considered the uncertainty of orthorectification and the uncertainty of margin delineation by
individual operator (±3.5%). The former was calculated using the buffer method with the width
equal to the half of the KH-9 pixel of 7.6 m. The combined uncertainty was ±4.3%.
The uncertainty of changes in glacier area were calculated using the uncertainty of co-
registration of KH-9 and Landsat images and the uncertainty of glacier margin delineation. The



combined uncertainties were ±5.6% for the 1976 and 1992 images and ±5.8% for the 1976 and
2013 images.

**3.3 Meteorological data**
Monthly statistics for air temperature and precipitation from the Narynkol station (42º43´N; 80
º11´E; 1806 m a.s.l.) was used (Fig. 1; Table 1). The station is located in a wide valley (about 20
km in cross-section) of the River Tekes, which has west-east orientation. The station was
established in 1947 and moved twice: In 1953, it was moved by 50 m east and in 1975, it was
mover 500 m north-east of its original location (Aliyakbarova, 2004). Currently, the station is
located at the south-eastern edge of a village Narynkol in which one-storey buildings
predominate. The nearest buildings are positioned 50-70 m away from the station and their
heights do not exceed 8-13 m. Although there has been no assessment of urban heat island in
Narynkol, the heights of the buildings suggest that it should be low. The Tretyakov rain gauge
was introduced in 1951 replacing a Naphier rain gauge (Aliyakbarova, 2004). In this study, we
used temperature for the 1947-2015 period and precipitation for the 1952-2015 period.
Several statistical tests have been used to examine temporal variability in the temperature and
precipitation records. In addition to the widely applied linear trend analysis, the Cumulative Sum
Control Chart (CUSUM) test (Mansell, 2003) and Mann-Kendall sequential test (Sneyers, 1990)
were applied. Both tests are used to identify approximate time of beginning of a trend or change
points in time series.

**4 Results**
**4.1 Glacier change between 1992 and 2013**
In 1992, there were 118 glaciers in the study region with a combined map area of 121.4±9.2 km$^2$
and by 2013, their area declined to 105.0±5.5 km$^2$. The total area loss was 16.4±5.9 km$^2$ or



13.5±7.5%. The rate of area reduction was 0.78 km$^2$ a$^{-1}$ or 0.64% a$^{-1}$. Six glaciers separated and 8
glaciers disappeared.
Similar to other regions (e.g. Kutuzov and Shahgedanova, 2009; Narama et al., 2010; Xu et al.,
2015), larger glaciers lost smaller proportions of their areas (Fig. 4 and Table 3). The absolute
area loss by glaciers of 1-2 km$^2$ and 2-5 km$^2$ classes were higher than that of the glaciers in 0.01-
1 km$^2$ class. However, due to a large number of glaciers in the 0.01-1 km$^2$ class, their combined
area loss was the highest (Table 3). In 1992, the smallest glaciers occupied 21.2% of the total
glacierized area and in 2013, they accounted for 52.8% of area loss. All glaciers, which melted
completely, ranged in size between 0.02 km$^2$ and 0.19 km$^2$. The largest glaciers (>5 km$^2$)
accounted for 55.9% the total glacierized area in 1992 and in 2013, they accounted for 20.9% of
total area loss. However, the absolute area loss by the six largest glaciers was relatively small
and close to the uncertainty of measurement.
The largest glaciers belonged to the compound-valley type (Table 4; Fig. 5). Three of these
glaciers (Mramornaya Stena, Simonov and No 104) are located on the northern slope of the
Saryjaz and in the Meridional Ridge respectively. The accumulation zones of these glaciers are
positioned at higher elevations reaching 4400 – 6150 m a.s.l. Tongues of three largest glacier –
Marmornaya Stena, Bayankol and Simonov – have an extensive debris cover which slows down
their retreat (Fig. 7 further in the text). Their low recession rates are consistent with the slow
wastage reported by Osmanov et al. (2013) for glaciers of similar size in the southern sector of
the Saryjaz. The largest absolute loss characterised valley glaciers (Table 4; Fig. 5). Cirque
glaciers exhibited higher relative loss despite their shaded positions. This was probably due to
their smaller areas which averaged 0.21 km$^2$ and location at lower elevations between 3440 m
and 4500 m in contrast to the valley glaciers which averaged 2.1 km$^2$ extending from 3550 m to
5840 m. The largest relative loss was characteristic of ice aprons followed by the flat-summit
glaciers. Although there are only three glaciers of these types in the study area, the high rates of
their recession are consistent with the trend reported by Kutuzov and Shahgedanova (2009) for





the Terskey-Alatoo and can be attributed not only to their small size but also to the fact that
glaciers of these types have large marginal areas and recession occurs along the whole margin
rather than relatively narrow glacier terminus.
Most glaciers in the region have northern aspect. Out of 118 glaciers, 55 faced north, 18 north-
west and 20 north-east and this is why the combined area loss was highest for the glaciers with
northern aspect accounting for 8.5±2.3 km$^2$. Glaciers with southern and eastern aspect, of which
there are only fourteen, lost the highest proportions of their area (Fig. 6).

### 292    4.2 Glacier change between 1976, 1992 and 2013

The combined area of 28 glaciers measured from the KH-9 Hexagon image from 1976 was
80.1±3.0 km$^2$ (Table 5 a). Glaciers in this sample were larger than on average across the region
with a mean area of 2.86 km$^2$. By 1992, their combined area had decreased to 73.5±4.7 km$^2$
(60.5% of the total glacier area in the basin) and by 2013, to 67.3±3.1 km$^2$ (64% of the total
glacier area in the basin). The rates of glacier wastage are shown in Table 5 b. Figure 7 illustrates
the recession of three large glaciers in this region while Figure 3 illustrates changes in areas of
smaller glaciers. The recession rates were slightly higher in 1976 - 1992 period than in 1992-
2013 although the differences are close to the measurement uncertainty.

### 302    4.3 Changes in temperature and precipitation

According to the climatic data from the Narynkol station, positive trends in temperature
significant at 0.05 level were observed in all seasons (Fig. 8; Table 1). The strongest increase
occurred in autumn and winter. The strongest trend of 0.58°C/10 a$^{-1}$ was observed in November
while in October and December and February temperature increased at a rate of approximately
0.30-0.45°C/10 a$^{-1}$. In January, when the Siberian anticyclone is at its strongest, the trend was
weaker at 0.20°C/10 a$^{-1}$. In September, temperature increase occurred at a rate of 0.23°C/10 a$^{-1}$.
The summer temperature time series was characterised by weaker interannual variability than



other seasons as indicated by the lowest value of the coefficient of variation (CV). The
application of the CUSUM and Mann-Kendall sequential tests confirmed the presence of a
continuous positive trend without significant abrupt changes in all time series.
None of the precipitation time series including annual, those for the standard meteorological
seasons or for glacier mass balance seasons (defined as September-May and JJA) exhibited
linear trend significant at 0.05 level. Both CUSUM and Mann-Kendell sequential tests indicate
presence of the opposite trends in the annual precipitation time series before and after 1976-1977
resulting from changes in the summer and spring precipitation (Fig. 9; Table 1). Strong negative
anomalies in annual precipitation were observed in 1975-1977. The 1975 annual total was two
standard deviations below the record mean, while in 1977 the annual precipitation values were
close to this threshold. Following the reversal of the negative trend, annual precipitation totals
were increasing until 1993, however, a number of dry years occurred in the 2010s and in
particular, 2012 was the second driest year on record with the annual total of 277 mm (Fig. 9).
Similarly to the dry period of the late 1970s, the decline in annual totals in the 2010s was due to
the reduction in spring precipitation. Both the dry periods of the late 1970s and 2012-2014
coincided with the period of positive anomalies in summer temperature. There is no statistically
significant difference between 1952-1977 and 1977-2015 annual and seasonal precipitation totals
(Table 1). There is also no statistically significant difference between precipitation averaged over
1976-1992 and 1992-2013 periods, 363±57 mm and 404±77 mm respectively, over which the
retreat rates of 28 glaciers were measured.

**5 Discussion**
**5.1. Glacier change between 1992 and 2013**
Glaciers in the Kazakhstani sector of the Tekes River basin have lost 13.5±7.5% of their area
over the 1992 - 2013 period retreating at a rate of 0.60 ± 0.3 % $a^{-1}$. In comparison with the
changes observed in other glacierized regions of the northern Tien Shan in approximately the



same time period, glacier recession in the Tekes basin proceeded at a slower rate. In the recent
assessment by Severskiy et al (2016), the retreat rate in the Zailiyskiy Alatau in the 1990-2008
period was reported as 0.89 % a$^{-1}$ while in the Djungarskiy Alatau the retreat rate was even
higher at 1.1 % a$^{-1}$ between 1990 and 2011. This difference can be attributed to the different size
of glaciers. In the Tekes basin, areas of three large glaciers with extensive debris cover
(Mramornaya Stena, Simonov and Bayankol; Fig. 7), accounting for 40% and 45% of the
combined glaciated area in 1992 and 2013 respectively, did not change beyond the error of
measurement (2.5±5.0%). This is similar to the Saryjaz Range, located south of the Tekes basin,
where glaciers, which are larger and positioned at higher elevation, retreated at 0.19% a$^{-1}$ leading
to the overall reduction by 3.7±2.7% in the 1990-2010 period (Osmonov et al., 2013; Table 1A).
The combined area of all other glaciers, excluding the three largest, in the Tekes basin declined
by 20.8±7.5 % or 0.99±0.36 % a$^{-1}$. These statistics are comparable with the results by Severskiy
et al. (2016) for other regions of the northern Tien Shan most of which feature smaller glaciers.
The mean retreat rates for all glaciers in the sample is very close to 0.57 % a$^{-1}$ reported by
Narama et al. (2010) for the central and northern Tien Shan for a shorter period of 2000-2008
(Table 1A).
The importance of the impact of debris cover on the retreat rates of glaciers can be illustrated by
a comparison of two individual glaciers of a similar size, type and aspect: Bayankol (6.2 km$^2$)
which has extensive debris and Sauruksaiskiy (6.8 km$^2$) which has a clear snout. While the area
of Bayankol declined by 2.0±5.0%, the area Sauruksaiskiy declined by 11.0±5.2%. Extensive
debris cover on the glaciers of the Saryjaz was identified by Osmonov et al. (2013) as one of the
factors predetermining their slow retreat. Pieczonka and Bolch (2105) also highlighted lower
retreat rates of debris-covered glaciers in the Kokashal-Too, Tomur and Inylchek regions in
comparison with other glaciers in these regions.
As in many other regional studies (Kutuzov and Shahgedanova, 2009; Narama et al., 2010),
smaller glaciers retreated faster and lost higher proportions of their area. Changes in the extent of



the largest glaciers with areas in excess of 5 km$^2$ between 1992 and 2013 were close the
uncertainty of measurements (Table 3) and changes in the extent of three largest compound-
valley glaciers were insignificant (Table 4). In contrast to other studies (e.g. Kutuzov and
Shahgedanova, 2009), cirque glaciers lost smaller proportions of their area than valley glaciers
but this is probably due to position of the accumulation zones of valley glaciers at higher
elevations.

**5.2. Glacier change between 1956 / 1976 and 2013**
A sub-sample of 28 glaciers was used to assess temporal changes in glacier recession rates. This
sub-sample included larger than average glaciers whose relative area loss was lower at 8.4 ±
5.9% than the regional value in the 1992-2013. Between 1976 and 2013, these glaciers retreated
at a rate of 0.43±0.16 % a$^{-1}$ and lost 16.0±5.8% of their combined area. Analysis of retreat rates
between 1976 – 1992 (0.52±0.35% a$^{-1}$) and 1992 – 2013 (0.40±0.28% a$^{-1}$) indicated that there
was no acceleration in the glacier recession rates and that the temporal changes in the recession
rates are within the uncertainty of the measurements (Table 5b). This result is consistent with the
low temporal variability in the glacier recession rates reported by Severskiy et al. (2016) for the
Djungarskiy and Zailiyskiy Alatau and contrasts the conclusions by Narama et al. (2010) who
reported a slight acceleration in glacier retreat rates in central and northern Tien Shan.
According to the Catalogue of Glaciers of the USSR (Vilesov, 1969), the combined area of the
same 28 glaciers was 86.3 km$^2$ in 1956 and, therefore between 1956 and 1976, these glaciers
were retreating at a rate of 0.36 % a$^{-1}$ (Table 5b). Uncertainty analysis is not possible with regard
to the 1956 data, however, the 1956-1976 retreat rate is close to those observed in the following
decades and the difference appears to be close the uncertainty of measurement of glacierretreat
using satellite imagery. Between 1956 and 2013, the glaciers lost 22.0% of their combined area
retreating at a rate of 0.39 % a$^{-1}$. This is very close the 0.37±0.22 % a$^{-1}$ retreat rate reported by
for the Chinese sector of the Tekes basin for the 1960 – 2009 period by Xu et al. (2015).




### 5.3. Comparison of the 1976 data with the data published in the Catalogue of Glaciers of the USSR

The Catalogue of Glaciers of the USSR (Vilesov et al., 1969) presented results from a large scale inventory conducted in the Tien Shan in the 1950s-1960s which are often used as the benchmark data in the evaluation of glacier wastage (e.g. Vilesov et al., 2006; Bolch et al., 2007, Severskiy et al., 2016). A direct assessment of the accuracy of the Catalogue data by Shahgedanova et al. (2010) for the Altai Mountains by re-mapping a sample of glaciers on the aerial photographs used in the compilation of the Catalogue of Glaciers (Dushkin, 1974) indicated that the combined glacier area published in the Catalogue was 5.5% higher than the re-mapped area and although for individual glaciers did not exceed 12%. By contrast, an assessment for the Kodar Mountains, eastern Siberia by Stokes et al. (2013) revealed that over 50% difference existed between areas of individual glaciers presented in the Catalogue of Glaciers (Novikova and Grinsberg, 1972) and the re-mapped data and a number of glaciers were missing from the Catalogue. While similar re-assessment is impossible for the northern Tien Shan including the Tekes basin because the aerial photographs were not preserved, the consistent rate of change between 1956-1976 and the later time periods, which is in line with temperature change (section 5.4), indicates that the Catalogue data are unlikely to contain major error with regard to the combined area of glaciers. However, within the area covered by the KH9 Hexagon imagery, one glacier with area of 0.6 km$^2$ as in 1976 was missing from the Catalogue of Glaciers (Vilesov, 1969).

### 5.4. Comparison with the RGI5.0 / GAMDAM

A comparison of our results with those published by RGI5.0 using data from the GAMDAM inventory (Nuimura et al., 2015) indicated that glacier area in the Tekes basin was underestimated by RGI5.0 / GAMDAM in comparison with this study. The methodology





adopted by Nuimura et al. (2015) explicitly excludes glaciers located on the slopes with gradient
exceeding 40º which are considered to be steep headwalls without permanent ice cover. While
this approach may be justified in other regions of the High Asia, in the Tekes basin, it results in
the exclusion of the accumulation zone of glaciers which were considered a part of glacierized
area both in this study and in the Catalogue of Glaciers (Vilesov et al., 1969). It also makes
comparisons with other regions problematic as none of the regional glacier inventories in the
Tien Shan (Table 1A) use this methodology.
Figure 10 illustrates the discrepancy between our analysis and RGI5.0 / GAMDAM data. The
combined map area of glaciers 89 (Simonov), 90, 91 (Bayankol) and 94 (Mramornaya Stena)
was 34.7 km$^2$ in 1999 – 2003 according to the RGI5.0 / GAMDAM while smaller glaciers No 92
and 93 are not accounted for. According to our measurements, the combined area of these six
glaciers was 47.4 ± 1.9 km$^2$ in 2013 after the removal of rock outcrops. The difference of 12.7
km$^2$ constitutes 12.1% of the combined glacier area in the Tekes basin in 2013. Examination of
the Landsat and high-resolution SPOT imagery (Figure 10) shows that crevasses indicating the
presence of ice were notable in the accumulation zones of glaciers (e.g. Glacier N 94) excluded
by RGI5.0 / GAMDAM. While there were no radar surveys of ice thickness in the accumulation
zones of glaciers in the Tekes basin, the radar surveys on the Sary-Tor glacier in the Ak-Shirak
massif (Petrakov et al., 2014) showed that ice thickness on the very steep slopes was 20-40 m.
Similar values were obtained by Kuzmichonok et al. (1992) for the Abramov glacier.
In addition, 59 small glaciers (including those separated from the larger glaciers) with a
combined area of 11.8 km$^2$ are not accounted for in RGI5.0 / GAMDAM. Overall, the combined
glacier area presented in RGI5.0 / GAMDAM for 1999 – 2003 is lower by approximately 25 km$^2$
or 24% than the combined area a decade later in 2013.

**5.5. Changes in temperature and precipitation.**



As in many other regions of the Tien Shan (Farinotti et al., 2015), increasing summer
temperatures appear to be the main driver of glacier retreat. The JJA temperatures have been
increasing steadily at a rate of 0.18°C per 10 years in the study area since 1953 (Fig. 8) with
linear trends explaining 25% of the total variance (Table 1). Correlation analysis has not revealed
any significant impacts of either global (e.g. El Nino – Southern Oscillation) or Northern
Hemisphere (Panagiotopoulos et al., 2002) teleconnections on temperature. The rate of change is
consistent with changes in summer temperature observed in other regions of the Tien-Shan
(Aizen et al., 1997; Kutuzov and Shahgedanova, 2009; Osmonov et al., 2013; Pieczonka and
Bolch, 2015). The strongest warming was observed in the Tekes basin in autumn and winter and
similar results were reported by Duethmann et al. (2015) for the Aksu River basin. Of all autumn
months, the highest rate of warming occurred in November (0.58°C per 10 years) and the lowest
in September (0.23°C per 10 a$^{-1}$). Annual temperature was increasing at a rate of 0.30°C per 10
years and this is consistent with 0.34°C per 10 years warming reported by Wang et al. (2011) for
the Chinese sector of the Ile River basin.
The observed warming is likely to lead to an increase in the proportion of liquid precipitation
both, at higher elevations in summer when glaciers receive most of their nourishment, and in the
transitional months (May and September) in the future. These changes may reduce accumulation
and increase ablation through higher temperatures in the current ablation season and its extension
into the transitional months.
There were no statistically significant linear trends in precipitation in any season although the
records exhibited strong interannual variability. Similar results were reported for other regions
of the Tien Shan by Aizen et al. (1997), Bolch (2007), Kutuzov and Shahgedanova (2009) and
Sorg et al. (2012) while Krysanova et al. (2015) reported positive trends in April-September
precipitation in the Aksu River basin in Kyrgyzstan and China. Precipitation was declining in the
1952–1977 period, increasing between 1977 and 1993 and then again decreasing until 2015 with
a very dry year of 2012 (Fig. 9). However, to date this variability did not seem to affect glacier



465 recession rates significantly on the time scale of two decades as the retreat rates for 1976-1992

466 and 1992-2013 were close with the differences within the error of measurement (Table 5) most

467 likely because there was no statistically significant difference between the average precipitation

468 values for these periods. The use of finer time steps in the analysis of glacier change may be able

469 to detect an impact of precipitation variability. Thus Severskiy et al. (2016) showed that the

470 highest glacier recession rates were observed in the northern Tien Shan between 1975 and 1979,

471 a period that was the driest on record (Fig. 9), exceeding those in 1990-2008 by the factor of two.

472 More recently, strong negative anomalies in annual precipitation driven by reduction in the

473 spring snowfall were observed in 2012-2014. The combination of a continuous increase in

474 summer temperatures and negative anomalies in spring and summer precipitation are likely to

475 have accelerated glacier wastage in the Tekes basin in the 2010s similarly to its acceleration in

476 the mid-1970s.

477

**478 6 Conclusions**

479 In the Kazakhstani sector of the Tekes basin, glaciers lost 13.5±7.5% in the 1992-2013 period.

480 This retreat rate appears to be slower than in many other regions of the Tien Shan in the same

481 period because of the presence of several large glaciers, whose areas remained unchanged, in the

482 sample. There was no significant change in the recession rates over time. A small reduction in

483 the recession rates in 1992-2013 in comparison with 1976-1992 appears to be within the

484 accuracy of our measurements: $0.40\pm0.28\%$ $a^{-1}$ versus $0.52\pm0.35\%$ $a^{-1}$. A steady increase in

485 temperature is a driving factor of glacier recession. The observed variability in precipitation

486 appears not to have a strong impact on glacier recession rates averaged over 15-20 year periods

487 although the influence of precipitation changes may be better detected if glacier change is

488 assessed at a finer time step. Positive temperature trends were observed in spring and autumn

489 month with the particularly high warming rates in autumn. This warming is likely to result in the



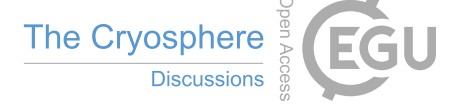

extension of the melting season and higher proportion of liquid precipitation leading to further
and potentially faster glacier recession in the future.

*Acknowledgements.* This work was conducted as a part of the project "Climate Change, Water
Resources and Food Security in Kazakhstan" funded by Newton - al-Farabi Fund (grant No

495    172722855).

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



Table 1. Characteristics of temporal variability in temperature and precipitation time series from
the Narynkol station. Values of $R^2$ referring to the linear trend in the time series significant at
0.05 level are highlighted in bold. σ is standard deviation. Trends for precipitation are presented
for two sub-periods because of the distinct opposite trends (Fig. 8).

| Season | Temperature (°C) | | | Precipitation (mm) | | | | | |
|---|---|---|---|---|---|---|---|---|---|
| | 1947-2015 | | | 1952-1977 | | | 1977-2015 | | |
| | Mean ±σ | °C/10 a | $R^2$ | Mean±σ | mm a$^{-1}$ | $R^2$ | Mean±σ | mm a$^{-1}$ | $R^2$ |
| DJF | -10.8±1.6 | +0.35 | **0.17** | 33±12 | +0.3 | 0.03 | 36±10 | +0.5 | 0.00 |
| MAM | 4.8±1.2 | +0.27 | **0.19** | 113±36 | -2.3 | **0.21** | 109±34 | -0.3 | 0.01 |
| JJA | 15.2±0.7 | +0.18 | **0.25** | 166±49 | -3.9 | **0.33** | 158±38 | +0.9 | 0.07 |
| SON | 4.2±1.3 | +0.39 | **0.36** | 80±23 | +0.3 | 0.01 | 84±30 | +0.6 | 0.06 |
| Annual | 3.3±0.8 | +0.30 | **0.54** | 393±78 | -5.5 | **0.27** | 387±72 | +1.3 | 0.04 |







Table 2. Details of the imagery used for glacier mapping.

| Satellite | Sensor | Path/ row | Spatial resolution (m) | Acquisition date |
|---|---|---|---|---|
| Landsat 8 | OLI TIRS | 147r030 | 30 / 15 (panchromatic) | 2013-09-09 |
| Landsat 8 | OLI TIRS | 147r031 | 30 / 15 (panchromatic) | 2013-09-09 |
| Landsat 5 | TM | 147r030 | 30 | 1992-07-13 |
| Landsat 5 | TM | 147r031 | 30 | 1992-07-13 |
| KH-9 Hexagon | | | 7.6 | 1976-01-12 |





Table 3. Area loss according to glacier size class.

| Glacier size (km²) | Number (1992) | Combined area (km²) | | Combined area change | | Average area change | |
|---|---|---|---|---|---|---|---|
| | | 1992 | 2013 | km² | % | km² | % |
| 0.01 – 1.0 | 97 | 25.6±3.4 | 17.0±1.5 | 8.7±1.3 | 33.8±7.9 | 0.09±0.01 | 37.8±7.9 |
| 1 - 2 | 10 | 12.8±1.2 | 10.3±0.7 | 2.4±0.6 | 19.0±5.5 | 0.24±0.06 | 19.4±5.5 |
| 2 - 5 | 5 | 15.2±1.0 | 13.4±0.6 | 1.9±0.7 | 12.2±5.1 | 0.37±0.15 | 13.1±5.1 |
| >5 | 6 | 67.8±3.7 | 64.3±2.7 | 3.4±3.3 | 5.1±5.1 | 0.57±0.56 | 6.8±5.1 |





Table 4. Area loss according to glacier type.

| Glacier type | | 1992 | 1992 | 2013 | Area reduction | |
|---|---|---|---|---|---|---|
| | No | Average size (km$^2$) | Area | Area | km$^2$ | % |
| Compound - valley | 3 | 16.3 | 48.7±2.6 | 46.9±1.9 | 1.85±2.4 | 3.8±5.1 |
| Valley | 21 | 2.2 | 45.6±3.2 | 39.0±2.0 | 6.53±2.2 | 14.3±5.8 |
| Cirque-valley | 19 | 0.62 | 11.7±1.2 | 8.9±0.6 | 2.82±0.6 | 24.1±6.1 |
| Cirque | 23 | 0.21 | 4.8±0.7 | 2.7±0.3 | 2.13±0.2 | 43.9±7.4 |
| Hanging | 49 | 0.19 | 9.3±1.3 | 7.1±0.6 | 2.25±0.5 | 24.1±9.0 |
| Ice aprons | 2 | 0.52 | 1.0±0.1 | 0.3±0.04 | 0.74±0.05 | 70.9±7.4 |
| Flat-summit | 1 | 0.2 | 0.2±0.03 | 0.1±0.01 | 0.08±0.01 | 47.6±6.5 |





Table 5. The combined area of 28 glaciers mapped using KH9 imagery (a) and its reduction (b)
between 1956 and 2013. Data for 1956 are from the Catalogue of Glaciers of the USSR (Vilesov
et al., 1969).
(a).

| Year | 1956 | 1976 | 1992 | 2013 |
|---|---|---|---|---|
| Area, km$^2$ | 86,3 | 80,1±3,0 | 73,5±4,7 | 67,3±3,1 |


(b).

| Period | | Area reduction | |
|---|---|---|---|
| | | km$^2$ | % |
| 1956-1976 | Total | 6.2 | 7.14 |
| | Per year | 0.31 | 0.36 |
| 1976-1992 | Total | 6.7 ± 3.9 | 8.3 ± 5.6 |
| | Per year | 0.42 ± 0.24 | 0.52±0.35 |
| 1992-2013 | Total | 6.2 ± 3.6 | 8.4 ± 5.9 |
| | Per year | 0.30 ±0.17 | 0.40±0.28 |
| 1976-2013 | Total | 12.9 ± 3.8 | 16.0 ± 5.8 |
| | Per year | 0.35±0.10 | 0.43±0.16 |
| 1956-2013 | Total | 19.0 | 22.0 |
| | Per year | 0.33 | 0.39 |





**Figure captions**

Figure 1. Study area.

Figure 2. Temperature (1947-2015) and precipitation (1952-2015) climatology for the Narynkol station (Fig. 1).

Figure 3. An example of glacier outlines from (a) Landsat 8 OLI TIRS image from 2013 (black outlines); (b) Landsat 5 TM image from 1992 (red outlines); and (c) Hexagon KH-9 image from 1976 (yellow outlines).

Figure 4. Area reduction according to glacier size (as in 1992).

Figure 5. Average rate of glacier area recession for different type of glaciers between 1992 and 2013.

Figure 6. The combined area loss (a) ($km^2$) and average rate of area loss (b) (% $a^{-1}$) by glaciers with different aspects between 1992 and 2013.

Figure 7. Example of glacier changes between 1976 and 2013: Bayankol (91), Mramornaya Stena (94) and Simonov (89). Landsat OLI TIRS image is used as background.

Figure 8. Seasonal temperature ($^o$C) for the Narynkol station: (a) DJF; (b) MAM; (c) JJA; (d) SON. The straight solid lines show record means. Note that different scales are used for different seasons because of the large annual range.

Figure 9. Seasonal precipitation (mm) for the Narynkol station: (a) DJF; (b) MAM; (c) JJA; (d) SON; (e) annual total. The straight solid lines show record means. Note that different scales are used for different seasons because of the large annual range.

Figure 10. Comparison of glacier outlines derived in this study with glacier outlines presented in RGI5.0 / GAMDAM (Nuimura et al., 2015). Higher-resolution SPOT imagery from 2007 illustrates the presence of crevasses in the accumulation zone of the Mramornaya Stena (No. 94) glacier which confirm the presence of ice cover in the area excluded by RGI5.0 / GAMDAM inventory.





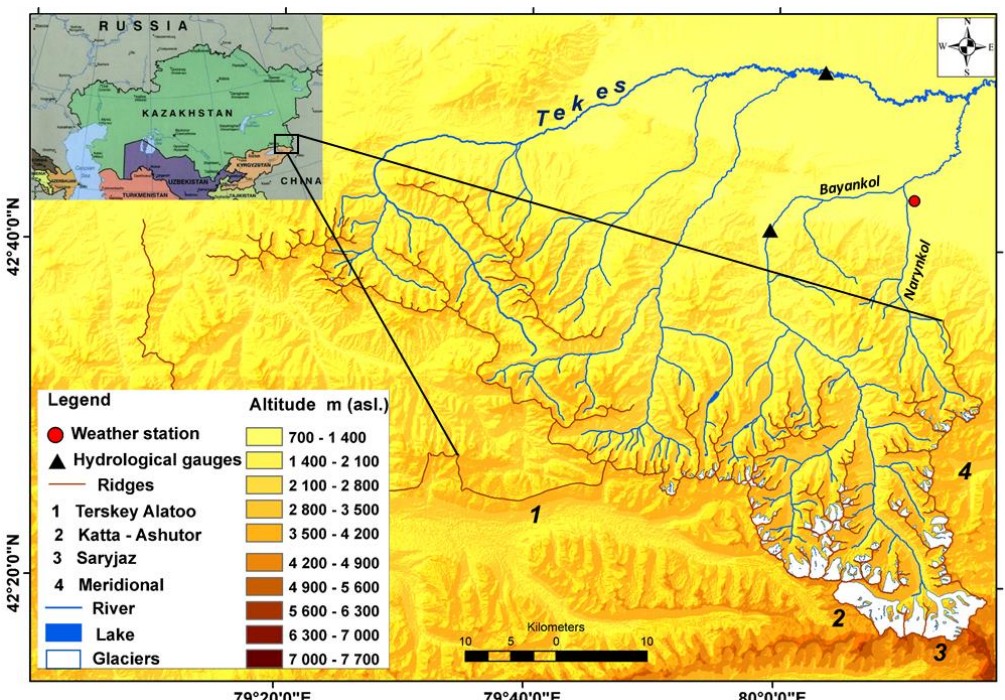



Figure 1. Study area.





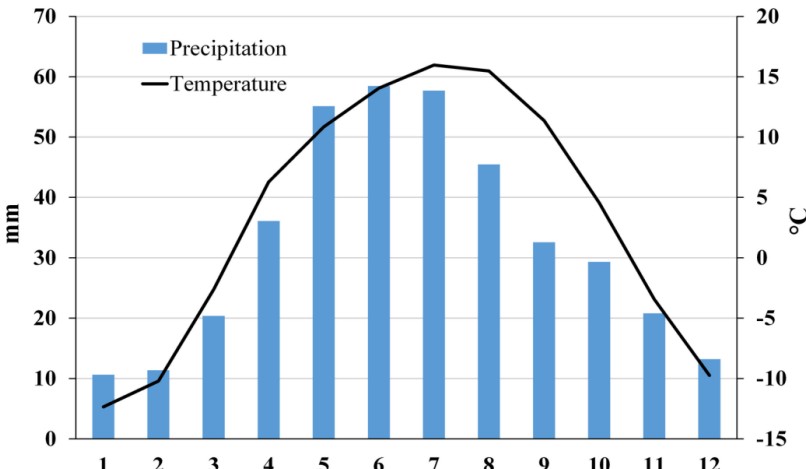



Figure 2. Temperature (1947-2015) and precipitation (1952-2015) climatology for the Narynkol
station (Fig. 1).



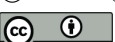

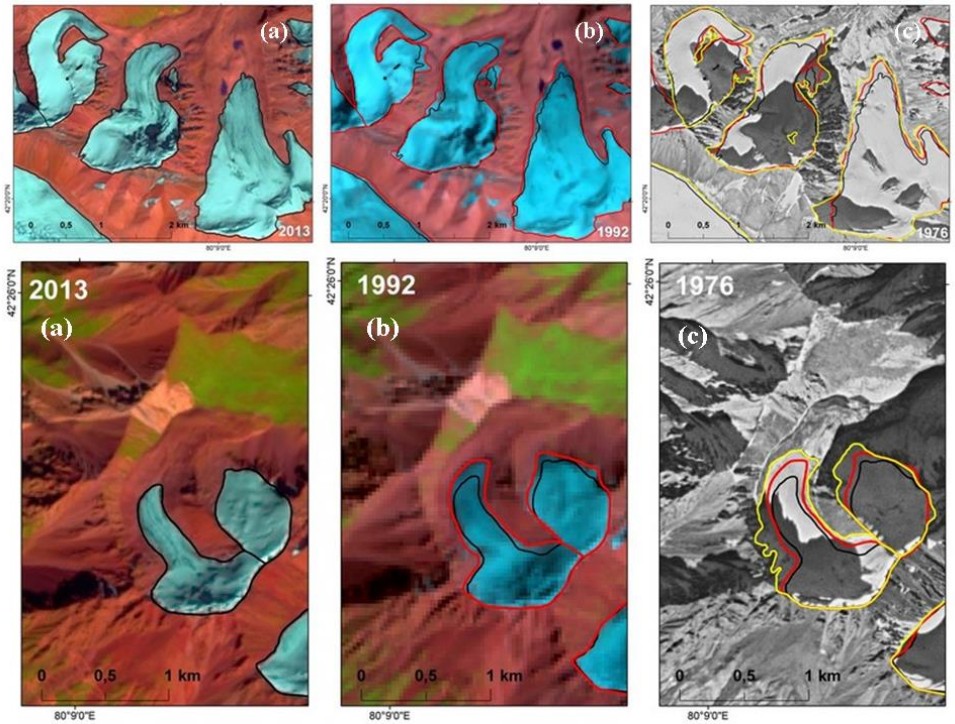

Figure 3. An example of glacier outlines from (a) Landsat 8 OLI TIRS image from 2013 (black outlines); (b) Landsat 5 TM image from 1992 (red outlines); and (c) Hexagon KH-9 image from 1976 (yellow outlines).





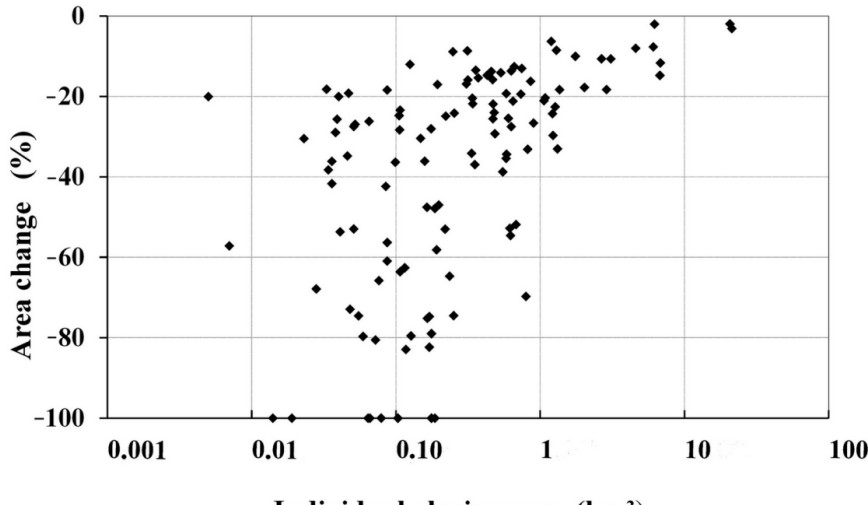



Figure 4. Area reduction according to glacier size (as in 1992).



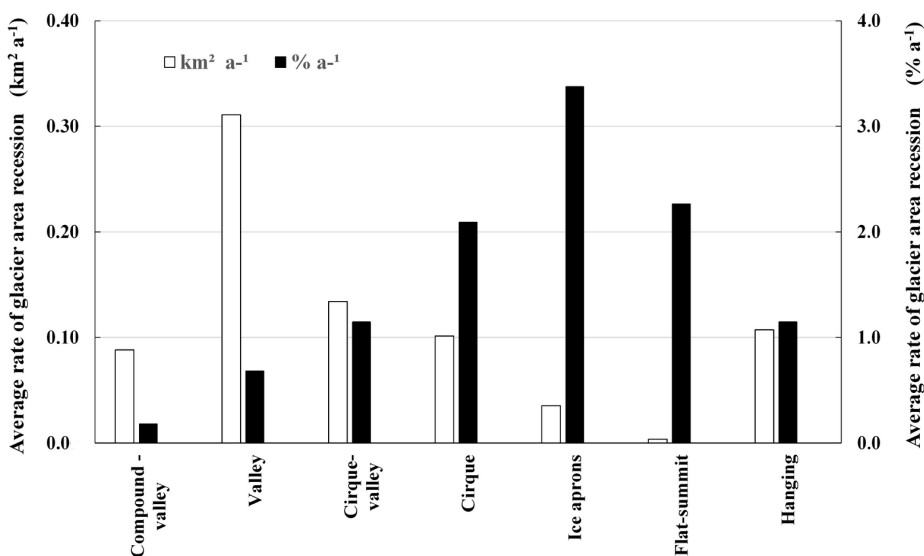



Figure 5. Average rate of glacier area recession for different type of glaciers between 1992 and
785    2013.




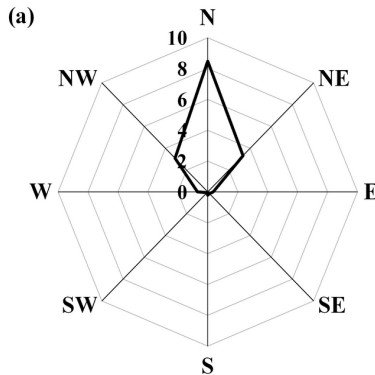
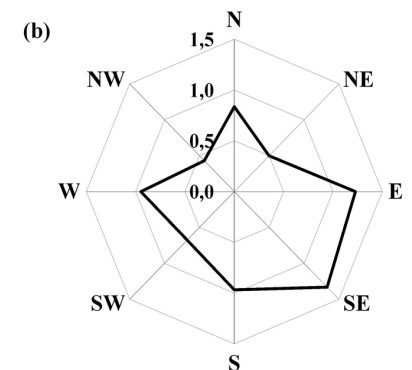



Figure 6. The combined area loss (a) (km$^2$) and average rate of area loss (b) (% a$^{-1}$) by glaciers
with different aspects between 1992 and 2013.




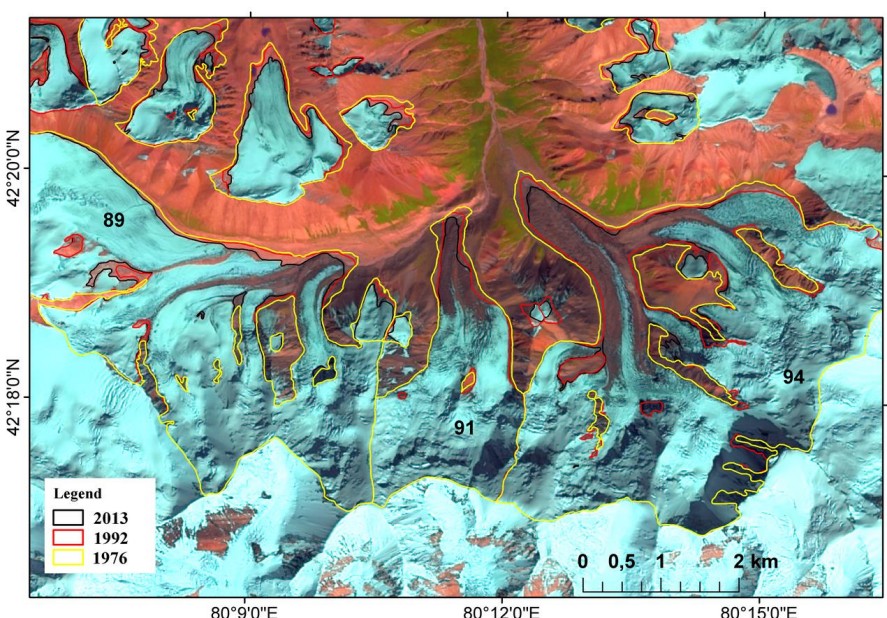



Figure 7. Example of glacier changes between 1976 and 2013: Bayankol (91), Mramornaya
Stena (94) and Simonov (89). Landsat OLI TIRS image is used as background.





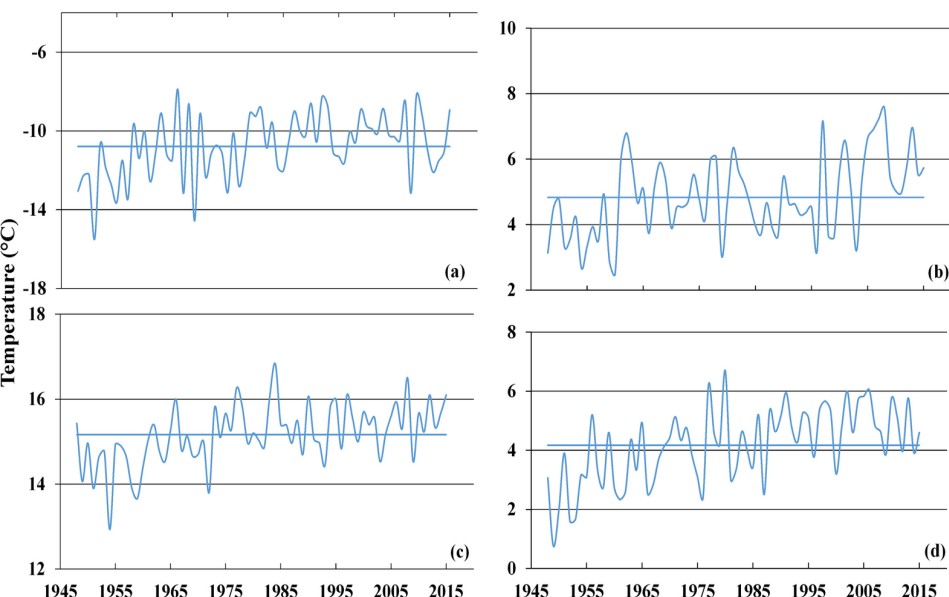



Figure 8. Seasonal temperature ($^{o}$C) for the Narynkol station: (a) DJF; (b) MAM; (c) JJA; (d)
SON. The straight solid lines show record means. Note that different scales are used for different
seasons because of the large annual range.



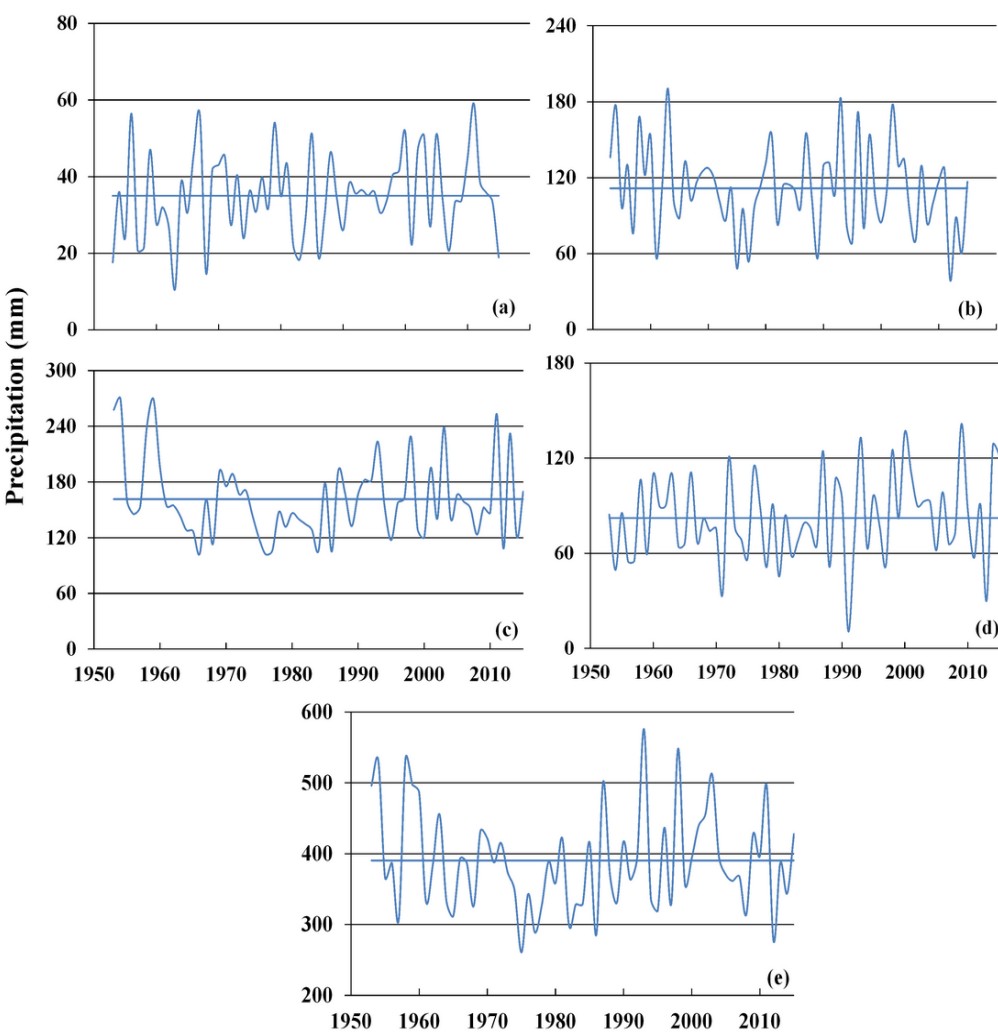

Figure 9. Seasonal precipitation (mm) for the Narynkol station: (a) DJF; (b) MAM; (c) JJA; (d)
SON; (e) Annual total. The straight solid lines show record means. Note that different scales are
used for different seasons because of the large annual range.





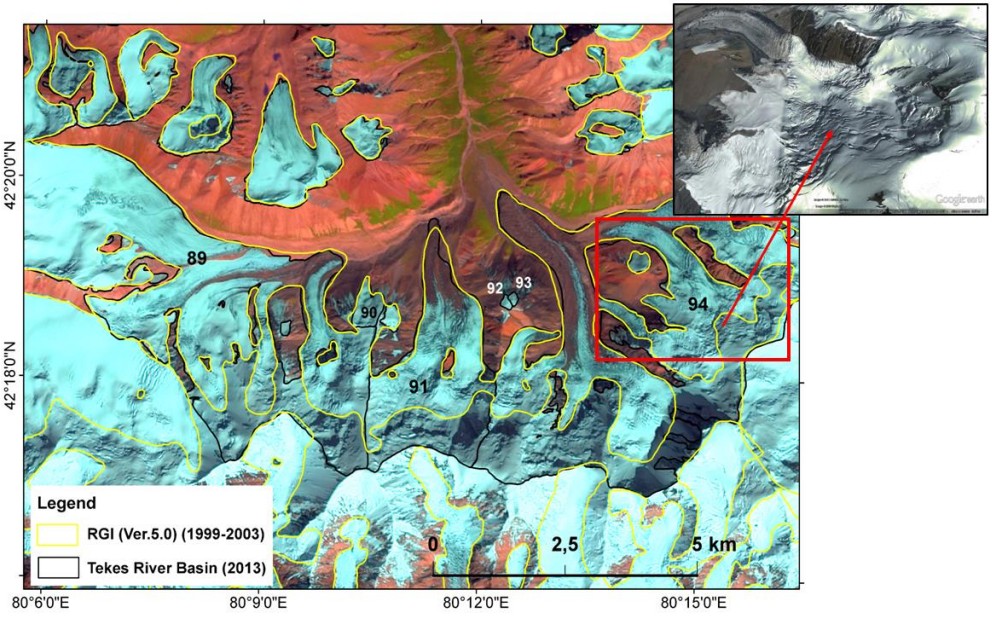



Figure 10. Comparison of glacier outlines derived in this study with glacier outlines presented in
RGI5.0 / GAMDAM (Nuimura et al., 2015). Higher-resolution SPOT imagery from 2007
illustrates the presence of crevasses in the accumulation zone of the Mramornaya Stena (No. 94)
glacier which confirm the presence of ice cover in the area excluded by RGI5.0 / GAMDAM
inventory.




Appendix A
Table A1. Results of assessments of glacier recession in Central Asia.

| Region | Period | Number/area of investigated glaciers | Surface area reduction (%) | Reference |
|---|---|---|---|---|
| 1 | 2 | 3 | 4 | 5 |
| **Northern Tien Shan** | | | | |
| Ala Archa basin | 1963–2003 | 48/36.31 km$^2$ in 2003 | 15.2 | Aizen et al., 2006, 2007 |
| | 1963–1981 | 42.83 km$^2$ in 1963 | 5.2 | |
| | 1981–2003 | 40.62 km$^2$ in 1981 | 10.6 | |
| Ala Archa Valley | 1964-2010 | 40.9 ± 1.8 km$^2$ in 1964 | 18.3 ± 5 | Bolch, 2015 |
| Northern slopes of Zailiyskiy Alatau | 1955-2008 | 307/287.3 km$^2$ in 1955 | 41 | Severskiy et al., 2016 |
| Upper Chon-Kemin | 1955–1999 | 31/38.5 km$^2$ in 1955 | 16.4 | Bolch, 2007 |
| Chon-Aksu | 1955–1999 | 48/62.8 km$^2$ in 1955 | 38.2 | |
| Sokoluk basin | 1963–2000 | 77/31.7 km$^2$ in 1963 | 28.0 | Niederer et al., 2007 |
| Ili River basin | 1960s - 2007/2009 | 2119/2002.94 ± 152.2 km$^2$ in 1960s | 24.2 ± 8.8 | Xu et al., 2015 |
| Jinghe River basin | 1964-2004 | 91/91.3 km2 in 1964 | 15.2 | Wang et al., 2014 |
| Sikeshu River basin | 1964-2004 | 150/114.6 km2 in 1964 | 15.4 | Wang et al., 2015 |
| **Central and Inner Tien Shan** | | | | |
| Akshiirak Massif | 1943-2003 | 178/371.6 km$^2$ in 2003 | 12.5 | Kuzmichenok, 1989; Aizen et al., 2006, 2007 |
| | 1977-2003 | 406.8 km$^2$ in 1977 | 8.7 | |
| | 1943-1977 | 424.7 km$^2$ in 1943 | 4.2 | |
| Ak-shirak Range | 1943-1977 | more than 170/436 km$^2$ in ~1950/60 | 3.0 | Khromova et al., 2003, Khalsa et al., 2004 |
| | 1977-2001 | | 20.0 | |
| Ak-Shirak Massif | ~1975 - ~2008 | 381 ± 15 km$^2$ in ~1975 | 8.8 ± 4.8 | Pieczonka and Bolch, 2015 |
| KokShal-Too | ~1975 - ~2008 | 587 ± 22 km$^2$ in ~1975 | 1.6 ± 4.9 | |
| Inylchek region | ~1975 - ~2008 | 1074 ± 41 km$^2$ in ~1975 | 3.0 ± 4.8 | |
| Tomur region | ~1975 - ~2008 | 964 ± 37 km$^2$ in ~1975 | 2.5 ± 4.8 | |
| Aksu Catchment | ~1975 - ~2008 | 3539 ± 135 km$^2$ in ~1975 | 3.6 ± 4.8 | |



| 1 | 2 | 3 | 4 | 5 |
|---|---|---|---|---|
| Sary-Tor Glacier (Ak-Shyirak Massif) | 1977-2003 | 1/3.54 km$^2$ in 1977 | 0.77 % a$^{-1}$ | Petrakov et al., 2014, Aizen et al., 2007 |
| | 1987-2003 | | 0.80 % a$^{-1}$ | |
| | 2003-2012 | | 0.67 % a$^{-1}$ | |
| Western Terskey Ala-Too | 1971–2002 | 269/245 km$^2$ in 1971 | 8.0 | Narama et al., 2006 |
| Eastern Terskey Ala-Too | LIA-2003 | 335/ 328 km$^2$ in 2003 | 19.0 | Kutuzov and Shahgedanova, 2009 |
| | 1965–2003 | 109/120 km$^2$ in 1965 | 12.6 | |
| | 1990–2003 | 335/328 km$^2$ in 2003 | 4.0 | |
| Big Naryn basin | the mid 20th century - 2007 | 700/614,5 km$^2$ in the mid-20$^{th}$ century | 23.4 | Hagg et al., 2013 |
| Naryn basin | the mid-1970s-mid-2000s | 1478/1210 ±30 km$^2$ in the mid-1970s | 23.0 | Kriegel et al., 2013 |
| Pskem | 1970-2000 | 525/219.8 km$^2$ in ~1970 | 19 | Narama et al., 2010 |
| | 2000-2007 | | 5 | |
| Ili-Kungöy | 1970-2000 | 735/672.2 km$^2$ in ~1970 | 12 | |
| | 2000-2007 | | 4 | |
| At-Bashy | 1970-2000 | 192/113.6 km$^2$ in ~1970 | 12 | |
| | 2000-2007 | | 4 | |
| SE-Fergana | 1970-2000 | 306/190.1 km$^2$ in ~1970 | 9 | |
| | 2000-2007 | | 0 | |
| Tarim Interior River basin | 1960/70-1999/2001 | 7665/17465.8 km$^2$ in 1960/70 | 3.3 | Shangguan et al., 2009 |
| Qingbingtan Glacier No.72, | 1964-2009 | 1/7.27 km$^2$ in 1964 | 21.5 | Puyu et al., 2013 |
| Qingbingtan Glacier No.74, | 1964-2009 | 1/9.55 km$^2$ in 1964 | 14.7 | |
| Keqikekuzibayi Glacier | 1964-2007 | 1/25.77 km$^2$ in 1964 | 6.8 | |
| Tomor Glacier | 1964-2009 | 1/310.14 km$^2$ in 1964 | 0.3 | |
| Qiongtailan Glacier | 1964-2003 | 1/165.38 km$^2$ in 1964 | 0.119 km$^2$ a$^{-1}$ | |
| Sary-Jaz River Basin | 1990-2010 | 1310/2055 ± 41.1 km$^2$ in 1990 | 3.7 ± 2.7 | Osmonov et al., 2013 |
| **Eastern Tien Shan** | | | | |
| Urumqi Glacier No. 1 | 1962–2009 | 2/1.646 km$^2$ in 2009 | 16.0 | P.Wang et al., 2014 |
| Middle Chinese Tien Shan | 1963–2000 | 70/48 km$^2$ in 2000 | 13.0 | Li et al.,2006 |



| 1 | 2 | 3 | 4 | 5 |
|---|---|---|---|---|
| Mt. Bogda region | 1962-2006 | 203/144.1 km$^2$ in 1962 | 21.6 | Li et al., 2011 |
| Mt. Harlik region | 1972-2005 | 75/98.3 km$^2$ in 1972 | 10.5 | |
| Mt. Karlik | 1977-2013 | 156/136.84 km$^2$ in 1977 | 21.9 | Du et al.,2014 |
| Karlik Shan | 1971/72 - 2001/02 | 122/126 ± 1 km2 in 1971/72 | 5.3 | Wang et al., 2009 |
| **Pamir** | | | | |
| Gissaro-Alay | 1957–1980 | 4287/2183 km$^2$ in 1957 | 15.6 | Shchetinnikov, 1998 |
| Pamir | 1957–1980 | 7071/7361 km$^2$ in 1957 | 10.5 | |
| Pamiro-Alay | 1957–1980 | 11358/9545 km$^2$ in 1957 | 12.5 | |
| Saukdara and Zulumart Ranges | 1978–1990 | 5/33.7 km$^2$in 2001 | 7.8 | Khromova et al., 2006 |
| | 1990–2001 | | 11.6 | |
| Muztag Ata and Konggur mountains | 1962/66–1999 | 302/835 km$^2$in 1962/66 | 7.9 | Shangguan et al., 2006 |
| Muksu River basin | 1980–2000 | −/468.4 km$^2$in 1980 | 7.4 | Desinov and Konovalov,2007 |
| Muztagh Ata | 1973-2013 | −/274.3 ± 10.6 km2 in 1973 | 0.6 ± 3.9 | Holzer, et al., 2015 |
| **Djungarskiy Alatau** | | | | |
| Southern Djungarskiy Alatau | 1956–2011 | 460/228.4 km$^2$ in 1956 | 47.4 | Severskiy, I. et al., 2016 |
| Northern Djungarskiy Alatau | 1956–2012 | 343/294,6 km$^2$ in 1956 | 38.4 | |
| Western Djungarskiy Alatau | 1956–2011 | 358/202.5 km$^2$ in 1956 | 44.1 | |
| Eastern Djungarskiy Alatau | 1956–2012 | 208/88.4 km$^2$in 1956 | 42.9 | |
| Karatal River basin | 1956-2012 | 285/199.2 km$^2$ in 1956 | 45.0 | Kaldybayev, et al., 2016 |
