# Peer review of "Assessment of Glacier Area Change in the Tekes River Basin, Central Tien Shan,"

_The Cryosphere, 2016_

## Referee Comment (RC1) · Anonymous Referee #1 · 18 Jul 2016

**Comments on "Assessment of Glacier Area Change in the Tekes River Basin, Central Tien Shan, Kazakhstan Between 1976 and 2013 using Landsat and KH-9 Imagery" by Usmanova *et al.*, submitted to *The Cryosphere*.**

**General Comments**

The manuscript submited by Usmanova et al. introduced the results of glacier change study in upstream basin of Tekes River, Central Tien Shan. This study used remote sensing and GIS techniques as the primary methods, and analyzed the climate change by obersevations of nearby meteological station. The results filled some gaps of glacier change studies over Tien Shan, and also contributed to the knowledge of regional climate change and its impacts on glaciers.

However, from my point of view, this manuscript is currently not suitable to be published on *The Cryosphere*, mainly due to several of its intrinsic defects. Papers about glacier change study that were recently published on *The Cryosphere* are either cover several aspects (e.g. mass balance, volume change, surface velocity, etc.) rather than glacier area change only, or cover larger regions and detailed in-depth glacier change analyses. Central Tien Shan is featured by larger continental glaciers, while this manuscript only focusing on a small partial of Tekes river basin with some small glaciers, and can very limitedly contribute to understandings of regional glacier change study, neither to the hydrological studies of Tekes or Ili River. The glacier change derived in the manuscript are also only relatively complete for the period of 1992-2013, while fairly partial in 1975-1992 (for only 28 glaciers) and uncertain in 1960s-1975 (by using unverified USSR glacier catalogue). The methods and data used in this manuscript are widely used by most researchers thus with limited innovations, while the results were also poorly presented. Therefore, **from the spatial coverage, the innovations, and the studying depth, this manuscript are all not sufficiently qualified to be published on *The Cryosphere***.

Besides above reasons, many other fatal shortcomings are still existing in the manuscript (see below), which are also significantly reduced the suitability of acceptance.

1. Currently the writing style of this manuscript is fairly awkward. The whole manuscript are glutted with typing and writing errors and/or mistakes, leads to relatively poor readability.

2. The quality of KH-9 image used in this study is also very limited, which was influced by heavy hill shade and large amounts of seasonal snow (see right). From my view, it cannot adequately supports the accurate delineation of glaciers in 1975.

[Figure]

3. The lack of glacier data on more dates has significantly lower down the effectiveness of the glacier change analyses in this study, which already awared by the authors. This seems blunder away more interesting glacier change patterns and the relationships to climate change.

4. The glaciologists of former Soviet Union have done great works to compile the Catalogue of Glaciers of the USSR, and we must give complete respects to their works. However, due to the source material and methods used during the compilation of the glacier catalogue, larger uncertainties must be introduced. That's why many researchers have performed validation study on the accuracy before using this glacier catalogue which resulted in a large range of uncertainties. This work is absolutely absent in this study.

5. The multi-stage change patterns of local climate can be easily read from the figures presented in the manuscript just by simple visual analyses. However, the authors only performed a linear trend analysis over

entire observation period, while ignored more meaningful climate change patterns at different stages. This somehow caused wastes of the valuable meteological observations.

**Some suggestions for the authors:**

1. Although the manuscript currently is not suitable to be published on *The Cryosphere*, it can be enough qualified to be published on other related international journals. Of course some shortcomings still need to be revised.

2. The national boundary should not became the barriers of glacier change study. if the authors still want to publish the manuscript on *The Cryosphere*, please consider to extend the study area to whole Tekes river basin, and use more efficient remote sensing data as suggested below, to retrieve detailed glacier change pattern.

3. According to my check, there are even higher resolution (1.8m) KH-4B imageries in 1970 on USGS website, with similar quality of current KH-9 image, but can cover all regions of the study area and neighbouring regions. Why not using those images and present a thorough study of glacier changes in whole basin?

4. A verification of the USSR glacier catalogure is strongly suggested to provide reference to its users to surrounding regions, whether the authors want to use it as a primary data source or not to study glacier change.

5. An in-depth check-up by native English speaker with sufficient glaciological knowledge and sufficient writing skills is strongly suggested.

**Specific Comments**

Line 20: "in the sample", use another word instead, e.g. "basin", and also for "combined" ("total"?).

Line 20-22: This should not be a finding of this study and thus better not to be listed here.

Line 24: "61% of … 1992" is not needed here, just one latest estimation is enough. The other and the comparison can barely tell anything.

Line 25-28: These two sentences actually tell same things, thus should better to be merged into one.

Line 30: This sentence should be split into two after "all seasons". Currently its structure is aweful, and the meaning is also inexplicable.

Line 31-34: The first sentence is suggested to be revised like "The precipitation show insignificant change patterns with strong fluctuations. It decreased in 1952-1977, increased in 1978-2000, and again decreased after 2000 with a number of dry years in 2010s". The second sentence makes no sense and should be removed.

Line 34-36: This conclusion (steady recession rate) is problematic. Also see comments on Line 485-488.

Line 40-41: Remove "at present". Besides, the RGI has become the new standard database of global galcier distribution, so the glacier data of Tien Shan should also cite the latest RGI database, i.e. RGI 5.0.

Line 41-44: "in the region" should be "in this region", and this long sentence should better to be split into two sentence around the "and recent assessments …". The "overall" should also be removed here, the "glacierized area and mass respectively" should better to be revised as "area and mass of glaciers, respectively, ".

Line 46-47: "potentially leading to …", this part of the sentence is aweful and need to be revised.

Line 47-49: The structure of this sentence is also aweful (with very confused subject-verb-object) and hard to follow. Please revise.

Line 49: "high retreat rates", maybe give an approximate value of the average change rate here will be better.

Line 52: What the "close to the accuracy of measurements" want to tell? It's confusing and also should better to be substituted by more specific approximate value of glacier change rate.

Line 53: Like Line 49 and 52, give a more specific approximate value of the glacier change rate rather than "similarly low recession rates" here. Besides, "similarly" should be "similar".

Line 54: "for the 1975-2008 period" should better to be revised as "during 1975-2008".

Line 55-56: "as did …", this part should be split into new sentence, and should be revised like "The study of Shangguan *et al.* (2009) gives similar rate during 1960s-2000 for glaciers in drainage basins of Chinese Tien Shan that flow to Tarim Interior River Basin".

Line 57: Remove "the" before glacier extent.

Line 58: "examining" should be "also examined", and add "which" before "highlight".

Line 59: Add "the studies of" before "Aizen et al."

Line 60: Revise "Having  analysed …" as "By using …" or "Depending on …".

Line 65-66: This sentence should better to be revised like "… showed that glacier recession rates in northern Tien Shan were comparable between the periods of 1955/56-1975 and 1990-2008, but 2-3 times higher during 1970s".

Line 67-69: Revise this sentence like "However, the value of glacier area for ealier stage in this study were simply read from Catalogue of Glaciers of the USSR that was compiled based on unarchived  historical aerial photography and topographic maps, made their uncertainty assessments problematic".

Line 70-73: Too much information contained in this long sentence, which makes it awkward. Please revise this long sentence, e.g. split it into multiple sentences or make the meaning and structure more clear.

Line 76-78: Same to Line 70-73, it's a too long sentence that contains too much information. Please revise it.

Line 80: Revise the last part of this sentence like "… to assess the changes of glacier extents of the basin".

Line 81: Revise as "… can dates back a century ago …".

Line 83: Add "(see Fig.1)" after "Saryjaz".

Line 87: Split the sentence into two at "and this was followed".

Line 88-89: Remove "More recently" because the contents after it (glacier changes) are not parallel or sequential to glacier distributions or inventories before. The following words should also be revised to more transitional words, e.g. "The glacier area in this basin has descreased 15.8% (0.45% a-1) according to Vilesov (2006)."

Line 91-92: "There are no …", it's incorrect because the study of Xu *et al.* (2015) extends to 2009 and can be a good reference for this basin due to its spatial neighborhood. Maybe what you want to say here is no glacier change study between 1990 and recent years for this basin.

Line 93-96: This part also talks about glacier inventories, thus it's better to be placed after USSR inventory (Line 88). Besides, in contrast to some former sentences, these two sentences should be merged into one. One suggested form is "The RGI 5.0 (Randolph Glacier Inventory, ……) contains updated glacier inventories for this region, which came from GAMDAM (……) glacier inventory compiled based on Landsat imageries acquired during 1999-2003".

Line 96-99: This sentence makes no sense and should better to be removed.

Line 101: Why didn't mention the glacier inventory of 1976 here? It should better than the inventories from Landsat imageries because of the higher resolution. Besides, remove repeated ")".

Line 103-104: Content of this point (glacier change) is repeated with previou one (ii).

Line 107-109: The longitude and latitude ranges are not necessary here which can be read from Figure 1 (or using more accurate ranges if you insist). "slope" is more common than "macroslope" in similar

literatures, and using "Ranges" rather than "Ridges" as well. The mountain names should be clearly marked on Figure 1 (also see comments on Figure 1).

Line 116&118: As commented on Figure 1, the glacier names mentioned should better to be marked on scale-enlarged map, to make clear which glaciers are the authors refering to.

Line 119-122: Using "total" rather than "combined" to represent the aggregated glacier area.

Line 123: What is "flat-summit glacier"? Does it means ice cap? Use ice cap instead if it is. "flat-summit glacier" is not a widely accepted definition (see GLIMS Glacier Classification Manual).

Line 172-174: I cannot imagine how the ice divides delineated from present ASTER GDEM can matching with the old USSR glacier catalogue. Please give a figure to illustrate it if possible.

Line 184: Use equivalent ground resolution instead of scan resolution here.

Line 188: Give the information of image that actually be refered (it should be one image rather that two or four images if using ERDAS Imagine 9.0).

Line 191-192: Were the accumulation zone margins for KH-9 image also mapped from Landsat imagery? Why didn't map it on KH-9 image? How to deal with different margins in accumulation zone like gradually emerged rocks?

Line 195: Comments on section 3.2

1) The authors should use more visualized forms, like formula, to describe what terms were involved and how the uncertainties were assessed.

2) From my view, the glacier area accuracies are mostly determined by the delineation processes rather than the orthorectification uncertainties. The residual error of image orthorectification mostly only affects the geolocation of glacier rather than their area (except the residual deformations in images on rugged terrain caused by errors in DEM or mis-placed GCPs), thus should better not included in glacier area uncertainty assessment in case of Landsat imageries because of their well-known find orthorectication accuracy (can be seen in a lot of literatures). However, it should be considered in case of KH-9 images because no equivalent resolution DEM and orthorectified higher resolution reference image can be found so far to accurately orthorectify them.

3) The mis-registration between images to derived glacier changes should be properly taken into account, especially in the case that same ice divides were used to split glaciers in different dates.

Line 198-199: Where did these 1:50000 maps came from? In which kind of coordinate system? How the transfromations between different coordinate systems were processed? Especially if the 1:50000 maps using coordinate system defined by former Soviet Union. Please clarify. Besides, I am very certain that the identifiable terrain features (they must be mountain peaks, river confluences, road crosses, etc.) picked from 1:50000 maps can not be always perfectly matching with those picked from Landsat imageries, because of the low spatial resolution of Landsat, and also the change in the features' geolocation with time. Therefore, the orthorectification uncertainty resulted from such method is inconvincible to me.

Line 200-201: How the uncertainties of glacier margin identification were defined for different types of images? Please clarify.

Line 202-205: What is the $RMSE_{x,y}$? The residual error of orthorectification? And how was it determined for different images?

Line 203-204: It is fairly unclear how the glacier area uncertainty was calculated here. According to your description, was it came from dividing the original area by the buffer extended area?

The percentage uncertainty is just one of the forms to present uncertainty. The normal description of glacier area uncertainty should using the form of x±y km², while how the value "y" was determined should be the key of uncertainty assessments. It is also the key in calculate percentage uncertainty (by y/x*100%).

Line 206-209: The 3.5% uncertainty of Paul et al. (2013) is a general result for large amount of glaciers. How did it been taken into account of the area uncertainty assessments in this study? By using simply root mean square?

From my view, the glacier area uncertainty should better to be presented in the form mentioned in above comment, and also for each single glacier mapped, rather than the total glacier area of the study region. Besides, the resulted 18.6% and 13% area uncertainty in 1992 and 2013 are too larger to study the glacier area change, because the uncertainty of change must reach to ~23%, which was much larger than the area change (13.5%) itself.

Line 210-211: Same as comments on this section, the co-registration uncertainty should better not to be considered as a part of the glacier area change uncertainty except the area uncertainties caused by using the common ice divides, which should be seperately calculated by proper way. The method suggested by Guo et al. (2015) may became an alternative way.

Guo, W., Liu, S., Xu, L., Wu, L., Shangguan, D., Yao, X., Wei, J., Bao, W., Yu, P., Liu, Q. and Jiang, Z. (2015). "The second Chinese glacier inventory: data, methods and results." Journal of Glaciology **61**(226): 357-372.

Line 211-214: The method to calculate glacier area change uncertainty is also problematic. Both glacier areas containing uncertainties, which certainly will be propagated into area change results. From my knowledge, the glacier area change uncertainty should be calculated based on the uncertainties in the glacier areas to calculate the change, rather than use a totally different method.

Line 215-224: The uncertainties in delineate debris-covered glaciers should not be so underestimated. Even by using high resolution Google Earth images, you cannot accurately identify where is the margins of debris-covered ice under many circumstances (e.g. as following Google Earth snapshots of debris-covered glaciers in the study region). So I suggest the authors to give a full consideration on the uncertainties of mapping debris-covered glaciers.

[Figure]

[Figure]

Line 229: The glacier mapping uncertainty from KH-9 imagery must be different from Landsat imageries because of the higher resolution, i.e. it should not be 3.5% anymore, and need to be properly re-evaluated.

Line 244: "… no urban heat island effects were assessed in …".

Line 247: Remove two "period".

Line 256 and below: How the absolute area uncertainty (x$\pm$y km$^2$) was calculated was not clearly described in section 3.2 (it is all about the relative uncertainties from my understanding). Please clearly specify it in corresponding section.

Line 257: Similar to comment on Line 211-214, the area change uncertainty should be calculated from area uncertainties by error propagation theory, which will result in $\pm$10.7 km$^2$ in this case, much larger than current $\pm$5.9 km$^2$.

Line 265-266: "The size of vanished glaciers ranging between 0.02 and 0.19 km$^2$".

Line 267-268: Replace "and in 2013, they" by ", however, they only", move "in 2013" to the end of this sentence.

Line 268-269: Remove this sentence.

Line 270-272: "The largest three glaciers, i.e. ……, which locating on the Meridional Range and the northern slope of Saryjaz, respectively, are all compound-valley type glacier". The first current sentence conflicts with Table 4, where only 3 glaciers belong to compound-valley type.

Line 272-273: Replace "positioned at higher elevatoin reaching" with "located in the elevation range of".

Line 273-277: "The tongues of these three glaciers are extensively covered by debris, which slows down their retreat rate (*here give their retreat rate*), and consistent with the rate (*it's better to give the reported rate here as well*) reported by Osmanov et al. (2013) in nearby Saryjaz basin".

Line 277-278: "The valley type glaciers exhibit the largest absolute area loss, and the cirque type glaciers also show higher relative loss despite their shaded locations (Table 4; Fig.5). The larger relative retreat of cirque glaciers was probably …".

Line 281-282: "The ice aprons have the largest relative area loss (…$\pm$…%) followed by the ice-caps (*please make sure that this is the correct terminology*) (…$\pm$…%).".

Line 287: Replace "aspect" with "orientation", and "faced" with "face".

Line 288-290: Breakup this sentence at "and this is why", replace the "combined area loss" with "absolute area loss", add "(Fig.6a)" at the end of this sentence, and add "By contrast, " ahead of next sentence. Replace the "(Fig.6)" with "(Fig.6b)".

Line 292: See general comment 2.

Line 305: Please clearly notation the corresponding abbreviations of periods in Figure 8 for autumn (SON?) and winter (DJF?).

Line 310-312: Where the "coefficient of variation (CV)" and the results for "CUSUM and Mann-Kendall sequential tests" are presented in the manuscript? Please clarify.

Line 313-315: Rewrite this sentence to make it structurally reasonable. Add "(see Fig.9)" at the tail.

Line 315-317: Similar to Line 310-312, where the results for "CUSUM and Mann-Kendall sequential tests" are presented?

Line 343-345: "This is similar to the southern neighbouring basin Saryjaz, where glaciers are relatively larger with higher elevation range, and retreated 3.7$\pm$2.7% during 1990-2010 (0.19% a$^{-1}$) (Osmonov et al., 2013; Table 1A)".

Line 346-348: "The area of all other glaciers except the three largest glaciers has declined by 20.8$\pm$7.5% (0.99$\pm$0.36% a$^{-1}$) during 1992-2013, and comparable to the results of Severskiy et al. (2016) for other regions of northern Tien Shan that mostly featured by smaller glaciers".

Line 349: Provide the value of "the mean retreat rate" in brackets.

Line 353-355: Add uncertainty to two area values in this sentence. See comments on Line 215-224, it's unreasonable that the clean-ice glacier has similar area change uncertainty with debris-covered glacier.

Line 362: Replace "in excess of" with "exceed" or "greater than", simply ">".

Line 363: I believe that the "uncertainty of measurements" will be much larger if using more rational uncertainty assessment methods.

Line 365-367: Double check this conclusion. The cirque glaciers have lost much higher proportions of their area than any type of valley glaciers, whatever in yours and Kutuzov and Shahgedanova (2009)!

Line 370-372: "These glaciers have a larger average size (… km$^2$).". Remove the words after "whose" because the comparison of the change rate between two different periods are meaningless.

Line 375: Use "a deceleration" rather than "no acceleration" here. It is very clear that the annual recession speed has decreased more than 1/5. Remove the words after "and that the temporal changes …" because it's very clear in Table 5 and this part makes no sense here.

Line 376-378: Replace the "low temporal variability in the" with "decelerated". Similar trend can be found in Severskiy et al. (2016), which the decelerations in annual rate after 2000 can be read in most regions.

Line 383-384: The part after "however" has less sense. The deduction of low measuring uncertainty according to similar change rate at different period is fairly speculative thus problematic. Actually, there was only very poor comparability between them. Besides, blank space was missed in "glacierretreat".

Line 385-387: "The retreat rate for the period of 1956-2013 is 22.0% (0.39% a$^{-1}$), which very close to the rate of 0.37$\pm$0.22% a$^{-1}$ in nearby Chinese sector of Tekes basin during 1960-2009 (Xu et al., 2015)."

Line 391-392: "…… presents a large scale glacier inventory in 1950s-1960s for Central Asian Tien Shan, which was often used ……".

Line 402-408: Similar to Line 383-384, the authors should be careful to give such speculative conclusion without any substantive evidence. The "consistent rate" theoretically can nearly prove nothing, neither to indicate the low uncertainty of USSR glacier catelogue, just as mentioned in last sentence. A more substantive and direct validation on the USSR glacier catelogue is strongly suggested, as discussed in General Comments.

Line 410: It is suggested to use GAMDAM only rather than RGI5.0/GAMDAM, also for other texts. The inclusion of GAMDAM in RGI 5.0 only need mentioned once in previous text.

Line 417-418: The reason for includes the steep headwall in glacier area should be clearly specified to provide a reference to other users of GAMDAM glacier inventory.

Line 418-420: It's ambiguous that what kinds of camparisons in whose studies are problematic. Please clarify it by clearly specify the problematic comparisons in corresponding studies.

Line 424-426: "… and 93 were absolutely not included. The area of these six glaciers was 47.4$\pm$1.9 km$^2$ in 2013 according to our measurements, 12.7 km$^2$ larger than GAMDAM inventory, which account for 12.1% of all glacier area in Tekes Basin in 2013."

Line 428-429: ", which excluded by GAMDAM inventory".

Line 434: "… total area of … not compiled in GAMDAM inventory".

Line 434-436: "As a whole, the glacier area in GAMDAM inventory in Tekes basin for 1999-2003 is 25 km$^2$ (24%) smaller than the area in 2013 of this study.".

Line 439: Add "e.g." before "Farinotti".

Line 441-442: Replace "10 years" with "decade" here, and for all other places. No related contents can be found in Table 1 for "linear trends explaining 25% of the total variance". Besides, what is the significance level of the trend of 0.18℃ per decade should also be mentioned. According to Fig.8c, the increasing trend is fairly insignificant after 1985. More phase-based trend analyses are suggested (also see comments on Fig.8).

Line 442-444: It seems that this sentence want talk about the driving factor of climate change. I suppose this study is focusing on glacier changes and their climate driving factors, rather than the climate change itself. So I suggest to remove this sentence.

Line 447-450: More phase-base analyses are needed for three other seasons. It is very clear in Fig.8 that the increasing trend of other three seasons also have stage differences.

Line 450: Use "decade" rather than "10 a$^{-1}$".

Line 453-455: The first sentence of this para is currently ambiguous. Increase in liquid precipitation of higher elevation and in "transitional months in the future"? How to understand it? Please specify it clearly.

Line 461: Breakup this sentence into two at "while", and replace it with "By contrast, " or "However, ".

Line 462-464: This sentence is conflict with the first sentence "no statistically significant linear trends". Please revise it to properly describe the insignificant trends.

Line 468-469: This is just one of the fatal shortcoming of this study. Also see General Comments.

Line 469-471: This result is questionable and worthy for more discussions in my opinion. The larger change rate (-1.67%/a) during 1975-1979 revealed in Severskiy et al. (2016) is too larger for glaciers with mean size of greater than 0.8 km$^2$, and actually it also need more conformation. Or give the value of the change rate as a source of visual comparison for readers if you insist.

Line 473-476: It need to be careful to draw such conclusion. The increasing trend of temperature after 2005 is fairly unclear for nearly all seasons according to Fig.8 (a panel for annaul change trend is needed as commented on Figure 8). There's also no data to support the acceleration wastage of glaciers after 2010s.

Line 485-488: It seems that you the authors have clear recognition on the largest shortcoming of this study. Please revise you study following this recognition, i.e. promote the time step of glacier inventories.

Line 488-491: From my view, the analyses on climate change are currently very shallow. The values of the precious meteological data have not been completely digged out. Some in-depth analyses are still required, e.g. change phases.

**Comments on References (In selective way):**

General comments:

    1) Hanging indents are needed for each literature;

    2) DOI is needed for every literatures.

Line 498-499: Use abbreviated journal name for "Journal of Climate".

Line 500-502: Use abbreviated journal name for "Annals of Glaciology".

Line 503-505: Use abbreviated journal name for "Global and Planetary Change".

Line 524-526: Use abbreviated journal name for "Journal of Applied Remote Sensing".

Line 535-537: Use abbreviated journal name for "Nature Geoscience".

Line 659-642: Wang is the family name, and Puyu is the given name, also for other authors.

Line 689-694: The two literatures of Wang et al. (2014) should be differentiated by 2014a and 2014b.

**Comments on Tables:**

Table 1: The contents of this table are largely repeated with Figure 8. It is suggested to label all the mean values, trends, and R$^2$ on Figure 8, and then remove this table.

Table 2: The information of KH-9 scene, e.g. the scene ID, is also need to be provided in the last row.

Table 3: An additional row of "Total" should better to be added in this table.

Table 5: The data in two sub-table need to be rearranged and better to be placed in one bigger multi-column table.

Table A1: The surface area reduction rates should better to be presented in annual. Besides, the results of current studies on the study area of this paper should be clearly presented for comparison.

**Comments on Figures:**

Figure 1: It's more intuitive to replace 1, 2, 3, and 4 with properly placed mountain name, and also for simplify the legend. Besides, the altitudinal legend seems not necessary because it fuzzily coupled with hill-shade effects. The scale of the map should be enlarged and the glacier IDs (or at least the IDs of glaciers mentioned in the text), and the glacier names as well, after minimize the legend. The country names should better to be marked on the main map.

Figure 2: More details are needed in this figure, e.g. the variation range or standard deviation of temperature and precipitation of every month as error bar. The data type or source should also be clearly marked as legends, e.g. "Mean precipitation (1952-2015)", "Mean temperature (1947-2015)".

Figure 4: The logarithmic units for horizontal axis should be clearly noted in the figure or axis caption.

Figure 5: Error bars are needed for each glacier types, both for absolute and relative changes

Figure 6: Two additional panels for glacier area and number distributions in different aspects are suggested to clearly illustrate the reason for different levels of glacier change.

Figure 7: See comments on Figure 10.

Figure 8:
1) A panel for annual temperature is needed, just similar to Figure 9.
2) The lines for linear trend of all phases should be added, with the regression formula and $R^2$.
3) It is obvious that the temperature for almost all seasons (except spring) kept relatively stable after 1975, which certainly will have influences on glacier change. Some phase-based trend analyses are thus strongly suggested to reveal more details about the temperature change. The results should also be marked on the figure.

Figure 9:
1) This figure can be merged into Figure 8 to compose a larger figure with two columns (left column for temperature while right for precipitation). This is better for the comparisons between the variations of temperature and precipitation.
2) Similar to Figure 8, the results of linear trend analyses should be marked on the figure.
3) More phase-based trend analyses are also suggested to give more details of the precipitation change.

Figure 10: This figure can be merged into Figure 7 by add the GAMDAM glacier outline into it. The yellow color for glacier outline is also unclear and should better to be changed.

---

## Referee Comment (RC2) · F. Paul (Referee) · 24 Jul 2016

**General comments**

The study by Usmanova et al. is presenting an analysis of glacier area changes in the Tekes Basin (Tien Shan) for 118 glaciers from 1992-2013 and a smaller subset of 28 glaciers from 1956 to 2013. They use Landsat data from 1992 and 2013, KH-9 imagery from 1976, the Russian glacier inventory from 1956 and outlines from the RGI for comparison. A detailed analysis of climate station data for the 1947/1952 to 2015 periods is presented as well. Particular emphasis is given to results from previous studies in nearby mountain ranges (that are compiled in an additional Table) and accuracy analysis. The study is well written (but I am not a native speaker), logically structured and some glacier outline overlays are provided as well. This gives overall a good impression. On the other hand, I also have a few larger objections that question the overall value of the study. The most important are:

(1) The scene from 1992 is in my opinion not suitable for glacier mapping, even if only glacier tongues are digitized manually. Most glaciers are completely snow covered and seasonal snow is hiding many glacier perimeters (see Fig. 1 in my Appendix). Using this scene would result in an overestimation of glacier area for many glaciers. In this regard I have a problem in matching the subset shown in Fig. 3b (top row) for 1992: There must be a massive snowfield in the lower right corner (where the '1992' is printed) but there is nothing (see Fig. 1 in my Appendix). Which leads to the question: Has this scene actually been used for the mapping or the one from 1989? Which brings me to my next point: Why has the poor scene from 1992 be used rather than the one from 22.8.1989? This scene is still not perfect but massively better than the one from 1992 (see my Appendix Figs. 1 and 2). Actually, the top row figure 3b (please use (a) to (f) next time) looks very much like the one from 1989. So is this now only a confusion of the figure by the authors (as they do not want to show the poor snow conditions in 1992) or has the 1989 scene actually been used (which might require to change all numbers)? Please note, for this point alone I would recommend rejection of the ms so that the authors have the possibility to clarify everything and perform the work properly.

(2) Furthermore, the glacier sample analysed here is really small (today's studies presenting change assessment typically cover some 1000+ glaciers) and biased towards a specific aspect sector. Whereas the latter restriction might be useful for a hydrologic analysis (or to reduce workload), such a restriction is close to senseless for area changes (and comparison to other regions), as these should always include the entire mountain range (considering all aspect sectors) to be comparable (e.g. glaciers exposed to the north might be larger and have thus a smaller area loss, see Fig. 6a/b). As the comparison to other studies is a central point of this study (Appendix A), the small and biased sample analysed here is not really suitable for this purpose. Hence, also this point puts a big question mark behind the usefulness of the study. I am aware that doing this analysis for the entire mountain range depicted on scene 147-031 (are there any further glaciers on 147-030?) would be a large amount of extra work, but otherwise the entire comparison part (which I like very much) would break down. Maybe results from colleagues can be included for a broader picture and a more useful analysis?

(3) Compared to my points (1) and (2), it is likely more easy to remove the detailed climatological analysis. Whereas this analysis is fine in itself, it has nothing to do with the here observed area changes, as glacier response times are not considered. I am aware that response times have also been neglected in other studies on area and length changes, but this does not mean it is a valid concept. And even for the 1D case (length change) the relation with climate

is complex and requires a (simple) glacier flow model to bring them together. For area changes the situation is even worse as the unknown ice thickness distribution plays a critical role for the observed changes (e.g. a glacier can be very flat or thick near the terminus). So please either skip the detailed climate analysis or look at time series of glacier mass balance (where the details would make sense). There is no problem in stating that the observed glacier shrinkage might be related to increasing temperatures (as in L439/440), but that's it.

(4) Finally, the accuracy analysis is strange and leads to very high uncertainties. It seems the authors have subtracted resulting glacier maps from each other to determine area change (what should never be done) so that geolocation plays a critical role in their assessment? The correct way of doing this would be to subtract the resulting (scalar) area values of each glacier polygon so that geolocation issues do not play a role. In consequence, all uncertainties given have to be revised and maybe the conclusions will then also be different. I also miss results of a multiple digitizing experiment to determine the real uncertainty of the derived outlines. This is really important when all outlines are based on manual digitizing. I give further details to this point in the specific comments below.

There are two smaller general issues requiring consideration:
(a) The area loss rate in $km^2/a$ has not really any meaning as it strongly depends on the total area covered and is thus incomparable among different regions and even within this region. I suggest removing these values throughout the entire ms and provide only relative area change rates (and the total glacier area at each point in time).
(b) The terminology should be more precise to avoid confusion. In my opinion the wording 'retreat'/'recession' (and advance) should be used when length changes are analysed (1D case). Change in glacier area (2D case) could be named area loss/gain or shrinkage/growth (for mass loss/gain: thinning/thickening). A consistent application of this terminology would also facilitate understanding of the text (e.g. in L47: why should runoff change when a terminus is retreating?). On P19 mass balance is frequently confused with area changes.

Overall and considering also my specific comments I would recommend rejection of the ms at this stage. This is basically to give the authors sufficient time to consider the comments, maybe redo larger parts of the calculations and write it up properly. If the changes obtained in this study were sensible and comparable with the others (presented in the Appendix), the study would be a welcome addition to the knowledge of glacier changes in the region.

**Specific comments**
L34-36: Though it is likely true that increasing temperatures are the reason for the shrinking glaciers, there is no way to relate a specific climatic forcing to glacier behaviour without considering response times (that might strongly vary from glacier to glacier and require consideration of the mass balance history). The details on temperature and precipitation (T/P) trends given in L30-34 suggest that details of glacier area change can be related to this variability. I suggest mention here only the more generalized conclusion about glacier shrinkage and increasing temperatures.
L74ff: There might be good reasons for selecting only this basin. But this study has a focus on comparing area changes across regions (Table 1 in the Appendix). For this purpose the selection of a drainage basin with an aspect bias gives incomparable and thus misleading results. For example, area changes can be smaller because glaciers exposed to the north are generally larger and smaller ones are better radiation protected. I thus recommend extending the analysis to the entire mountain range until all aspect sectors are included.

L90: Chinese sector: As before: I would remove (or mark) all studies from the comparison that potentially have aspect biases with likely impacts on the reported change rates. Otherwise a comparison makes little sense. If the data are available, one can also compare the changes per aspect sector and size class. At least it should be indicated in the Table, which studies suffer from such aspect biases due to drainage dive or country borders.

L119: Minimum elevation is not a good descriptor for the climatic regime. Can you give instead mid-point, mean or median elevation? These are closer to ELA and have thus a climatic meaning.

L119-123: These details about the glacier types are interesting but the differences are not fully clear to me (e.g. compound-valley, valley, cirque alley). If this differentiation is required for the later interpretation, can it be illustrated somehow (graphical/photos)?

L123: Are there any mass balance glaciers nearby or geodetic measurements that could be added here?

L134: What is the 'glacier tongue elevation'? Can this also be related to mean elevation (as a proxy for ELA)? This would be relevant for a climatic characterization of the glaciers.

L146: Nothing from Corona acquired in the 1960s here?

L155: 'at the end of the ablation period' does not mean that snow conditions are sufficiently good to map glaciers. This is only possible in years with a very negative mass balance and at best no seasonal snow left outside of glaciers. Otherwise the impact of the snow conditions on the mapping accuracy should be described as well. I think the OLI scene from 2013 can be used, but conditions are on the edge of being acceptable.

L157: 'suitable for glacier mapping': I disagree that this scene can be used. It only works for the lower parts of the largest tongues; all the smaller ones are snow covered. If this scene has really been used (Fig. 3b suggests something different, see above), please perform the analysis again with the scene from 22.8.1989 (see Fig. 2 in my Appendix). This scene is also not perfect, but much better than the one used here.

L160/1: Relative error also strongly increases towards smaller glaciers when manual digitizing is applied (just test it!), so this is not a good argument. The reason for using automated mapping as a first step is to get at least for the clean ice reproducible results. This applies in particular for the region presented here, where most of the glaciers are debris free and only very few (larger ones) have to be manually corrected due to debris. When all glaciers in scene 147-031 would have been mapped, the story is a different one. But also here the automated classification would provide a robust and reproducible first estimate of glacier coverage. Moreover, with this approach the (optically thick) percentage of debris cover can be determined for each glacier afterwards by grid subtraction. This information would be valuable for modellers (e.g. mass balance, future glacier development).

L166: Yes, and then you cannot use GAMDAM because it applies a different glacier definition. This is exactly the reason why automated mapping should always be applied as a first step (if a SWIR band is available). How does manual digitizing help when half of the glaciers are missing? Please note: as far as I know the GAMDAM inventory will be repeated using a more common definition of glaciers.

L169: Checking debris cover with HR images in Google Earth is fine, but where the available scenes suitable (e.g. regarding snow conditions)? Please describe this shortly.

L176: This is also a reason why automated clean ice mapping is helpful: rock outcrops are recognized and automatically excluded from the glacier map, independent of their size.

L188: Have both Landsat images been used to collect GCPs? As far as I know the higher resolution pan-chromatic sensor on ETM+ or OLI is preferably used for this. Please clarify.

L201ff: The Granshaw and Fountain (2006) study uses the buffer method to consider the thickness of a glacier outline on a map and the positional uncertainty of a map from 1958.

Both types of uncertainty do not really apply to the satellite images used here that are likely orthorectified with the same DEM and the same set of GCPs. But the key question is, why should geolocation uncertainty impact on glacier area? When glacier areas are determined independently for each glacier and the scalar values are subtracted to determine the change, where is the issue with geolocation? Unfortunately, it is not reported how area differences are determined here (please add!), but the description in L210-214 indicates that glacier grids have been subtracted after co-registration? If this is the case, please never do it this way as it blurs the quality of the results with the geolocation uncertainty.

When manual digitizing is performed, the - in my opinion - only useful way to determine uncertainty of the obtained glacier areas is from a multiple digitizing experiment (10-20 glaciers 3 to 5 times). This is a value specific to the analyst (and thus the study) that cannot be "taken" (L206) from another study or neglect the much higher uncertainty when it comes to the delineation of debris cover glaciers as mentioned in L216. The result of this exercise will be a size-dependent uncertainty envelop as shown in Fig. 3 of Granshaw and Fountain (2006). At best, size-class specific mean values of this uncertainty are used to obtain the correct value for the entire sample. Please also discuss the impacts of wrongly mapped glaciers due to seasonal snow hiding the glacier perimeter, in case it matters.

L231/2: As mentioned before, co-registration uncertainty should not play a role when area values are determined independently. What really counts is the uncertainty of the glacier margin delineation. And this should also be determined for some (isolated) glaciers on the Hexagon scenes. It would be interesting to see if these are lower than for Landsat due to the higher spatial resolution.

L236ff: As mentioned in the general comments, I think the detailed analysis of meteorological data is not required as a clear link is only given to mass balance measurements that are not presented here.

L256/7: The uncertainty given for the three numbers is now 7.5, 5.2 and 36%. Where do these numbers come from? I cannot link them to what is described in section 3.2.1. In particular the 36% uncertainty for the area change is strange. Why is it so large?

L274: I would not consider Simonov as having 'extensive debris cover'. There is some widening of the medial moraine near the terminus, but apart from this the ice is clean. The eastern tributary of Simonov, which is currently building the terminus, also seems to be very steep, with likely short response times. I doubt that its retreat is 'slowed down by the debris'. The same might apply to Bayankol, despite its more extensive debris cover.

L275/6: Terminology: by using here 'recession rates' and 'wastage' I would assume that length changes are compared to mass changes. Please use area loss rates when area loss rates are compared. In this regard, why can glacier changes in this region be compared to the one studied by Osmanov et al. (2013)?

L286: And due to their small elevation range they are likely much more sensitive to climate change (i.e. a small upward shift of the ELA).

L302: This is an interesting analysis of climatic variability, but the link to the observed area changes cannot be made. When precipitation variability is analysed over the same period as glacier change (L328/9), please look at mass balance time series (response times!).

L352ff: I think this comparison is free of scientific reasoning and not useful to demonstrate cause and effect. How should this work with two arbitrarily selected glaciers without considering flow velocities, surface ablation, ice thickness distribution and response times?

L357: slow retreat: does this mean here length, area or mass change? Are the glaciers in this region really comparable? They should at least be in the same size class.

L358: As long as area change rates are also a function of glacier size (e.g. Fig. 4 in this study) and debris-covered glaciers tend to be larger and longer, different retreat rates must not necessarily be a result of the debris cover, but can also be due to their larger size.

L370-376: This is basically a repetition of results. The key conclusion about the acceleration trend could also be provided in the results section. So I suggest deleting it here.

L378/9: When it is only a 'slight acceleration' I would not state that this is 'in contrast' to this study. The differences might be minor (or within the uncertainty).

L403/4: This is certainly a glaciological far stretch as it neglects everything we know about glacier response to climate change (see discussion above, I will not repeat it here).

L415: I think the point in excluding these areas was not that they are without permanent ice cover, but that these regions do not really contribute to mass loss and run-off. And this was the purpose for the inventory. Maybe better compare with the new Chinese Glacier Inventory (CGI2) as this would be more interesting.

L419/20: This likely applies to all inventories compiled elsewhere. This is also the reason why a new version of GAMDAM is currently in preparation (incl. accumulation areas).

L421ff: This is more a presentation of results rather than a discussion. But I would replace it anyway with a comparison to the CGI2. The differences to GAMDAM are well known in the mean time and are currently corrected (so the information is out dated soon).

L439-442: I think this generalized statement is fine, but I would not go beyond it.

L442-452: This is interesting in itself, but not required for the area changes presented here.

L453-457: This sounds very speculative. Are there any measurements to prove it?

L468/9: This is certainly true - when mass balance is analysed! When length changes are analysed, the time step should at least match the response time, for the area changes analysed here, do not expect to see anything, as non-climatic controls (ice thickness) take over.

L470: 'glacier recession rates': please never mix it up again: they look at mass balance!

L479/484: I expect the accuracy estimates and the conclusions to change with a correct accuracy assessment (and the values itself when the scene from 1989 is used).

L475: This might be true but it is not relevant here (as 'wastage' means mass balance).

L487: Yes, in case 'glacier change' means mass balance!

**Smaller issues**

L27/8: in the glacier recession => in glacier area loss

L28-30: I think this sentence is not required in the abstract.

L31: per 10 years => per decade

L39: glaciation => glacierization

L44/49/51: recession/retreat => area loss or shrinkage

L46: 'potentially ... lakes' => remove (changes in extent might not impact on run-off). Is formation of lakes an 'impact of glacier retreat'? I would argue that it is a consequence.

L135: explain JJA

L149: when cited here (even in brackets) it should be Fig. 3 rather than 9

L171: just write: "The ASTER GDEMv2 (http://...) ..."

L176: deduced: subtracted?

L187: Why OLI TIRS? Has the thermal infrared sensor been used as well? The optical land imager is just named OLI (without TIRS)

L261-264: Absolute area losses are not really comparable. I suggest deleting.

L337: retreat rate: I assume this is the area change rate rather than length change rate?

L342: glaciated should be glacierized for contemporary glaciers.

L352: retreat rates: this is like the area loss rate?

L357: (2105) is likely (2015)

L358: retreat rates: length or area changes?

L367: ... as well as many other factors such as special topographic conditions (shadowing, snow avalanching).

L382: retreating: losing area

L413: underestimated: I suggest writing 'lower' as this is more neutral.

**References**

Just as a remark: To increase access to the references, it is possible to use indentation from the second line on.

**Tables**

Below I comment on the tables as they are. These comments will mostly also apply for a new/revised version of the ms.

Table 1: I would suggest removing this table. Without analysing mass balance the investigation of seasonal trends in T/P is not relevant for this study.

Table 2: The Landsat 8 sensor used here is OLI, please remove TIRS. Please clarify in the main text why row 30 is also required. It is unclear from Fig. 1 which of the glaciers are not covered by 147-31. Please also give the image ID of the Hexagon scene used (DZB1211-500142L008001?).

Table 3: When the heading row gives area changes, all values miss a minus sign. It is unclear what 'Average area change' means. This has to be described in the caption. Is it the combined value divided by the number of glaciers? This works for the km$^2$ column, but what is here in the % column? I would also provide annual change rates, i.e. column 6 divided by 21 (or 24 when the better scene from 1989 is used). And please add a row with totals.

Table 4: There should be two further columns: per cent of the total area covered) between column 4 and 5) and the per cent area reduction per year (at the end). Here the values do not need a minus sign as 'Area reduction' is already including this. However, I suggest being consistent and using 'Area change' on top and add minus signs to all change values.

Table 5: a) Please use a dot instead of a comma (86.3) for all values and consider to just listing them in the main text or in the 5b) table. Overall, I think there is no need for the 5a) table. For b) I suggest removing the 'per year' line and add the value for the change rate as an extra column only for the per cent column. And please use area change on top and add minus signs.

**Figures**

Below I comment on the figures as they are. This might also be helpful for any future version of the study. In case the authors considerably extend the size of the study region, other/additional figures might be required.

Fig. 1:

The image shows major hydrological basins with glaciers squeezed in the lower right corner. As this study has a focus on glacier changes rather than hydrology, I suggest showing a close-up of the glaciers instead. Please also indicate the location of subsets on this map and show which glaciers are covered by the Hexagon scene (footprint). Annotations of specific glaciers could be indicated as well. Please note: there are many glaciers near the numbers 2, 3 and 4. When they are not shown without a note in the figure caption, the map is highly misleading. I suggest showing all glaciers in the region and highlight those being investigated (whatever the study region will be). The two long black lines are wrong by location and are not required. Please indicate in the figure caption what the subset and the black square on the subset should indicate. Figure caption: Please write a complete sentence (and give full details).

- Can the monochrome colour table of the background map be replaced with something more atlas like (starting with light green)? And Altitude in the legend should be elevation (and asl. should be a.s.l., but together with elevation a.s.l. is not required).
- The map looks like if there is a hill shade in the background, which question if the colours are only related to elevation. Please clarify and adjust the colours accordingly
- Where are the boundaries between the basins 1-4 and Tekes?
- I do not see any glaciers on the map that require scene 147-30. Are there any? If yes, please show the scene footprint (or the relevant part of it).
Fig. 2:
I think this figure can remain for the general climatologic characterization of the study region, but please add the elevation of the station, and that month is depicted on the x-axis. Caption: It is not only T/P that is shown: It is "Mean monthly temperature ...". Add annual mean T and annual sum of P in the caption.
Fig. 3:
Please use (a) to (f) when labelling 6 figures. The glacier in the lower row is very small compared to image size, please show a close-up. '(b) figures': please never use red outlines on a red background, white or light green might be better choices. I mentioned already above that the 1992 images cannot be from 1992, there is much more snow (see Fig. 1 in my Appendix).
Fig. 4:
I suggest using a font without serifs (such as Arial or Helvetica) for the labelling of all graphs. Please also add minor tick marks on both axes. Instead of 'Individual glacier area' I would write 'Glacier size in 1992'. The two dots smaller than 0.01 km$^2$ are likely snow patches rather than glaciers (that flow). I suggest using a lower limit of 0.01 km$^2$. The caption should read "Scatter plot of glacier size versus relative change in area from 1992-2013".
Fig. 5:
As mentioned above, absolute changes in glacier size have a very limited meaning. I think they can be shown here to illustrate the contrast. However, I think it would be more interesting (or maybe just add it) showing the contribution of each class to the total area change, maybe expressed as a ratio of their percentage to the total glacier area divided by their percentage of contribution to the change. This would indicate which glacier types contribute disproportional to the overall loss. Of course, such a graph would be more interesting when the sample size is larger.
Fig. 6:
Although a figure of the aspect distribution is not shown, this figure clearly illustrates a key problem of the study: The sample is strongly biased towards glaciers with a northerly exposition whereas the strongest area changes come from glaciers exposed to the SE. Mean area changes would likely be different without this bias.
Fig. 7:
Please do not use red outlines on a red background as this is difficult to see; maybe white, black and yellow works. There is limited change in the lower half of the picture. I suggest shifting the close-up to the north to show more of the study region or even go to portrait format and show the changes of the entire study region? Caption: Provide glacier names on Fig. 1. The sensor name is Landsat OLI. I suggest writing: 'The Landsat OLI image from 2013 is shown in the background.'
Fig. 8/9:
Please remove, there is no relation of this variability to the reported area changes.
Fig. 10:
I think we are all aware that the GAMDAM inventory has used a different interpretation of glacier extents. This will soon be changed (also in the RGI) so that the information is outdated from the beginning. I thus suggest removing the figure (or show them on top of Fig. 7, e.g. as a black-doted line). A close comparison with the CGI2 would be interesting instead.

Appendix Table
I like this overview very much but find the one to ten glacier samples not really comparable to those with a few hundred or even thousands of glaciers. In view of my critique on this study, I suggest either removing or at least marking those studies who have also an aspect bias or the other way round mark those who cover entire mountain ranges. For all studies I would add the area change rate per year in a further column, as only these values are comparable across regions. Maybe a graphical representation of the values listed in the table can be added as an overview (e.g. period vs. relative change rate with colour coded regions and line-style coded sample size (<100 dotted, 100-1000 dashed, >1000 solid))? This would give an easy access to the observed variability of change rates and the investigated time periods.

**My Appendix**

[Figure]

Figure 1: For point (1) of the general comments showing a subset of the study region. Left: The image in Fig. 3b of the study that should be from 1992 but is looking like the one from 1989 (middle). The real snow conditions in 1992 are shown to the right suggesting that the 1992 scene is indeed unsuitable for mapping glacier extents.

[Figure]

Fig. 2: Comparison of snow conditions: Left the scene from 1989 that should have been used, right the scene from 1992 that (maybe) has been used. The much more extensive snow cover (light blue) in the 1992 scene is easily recognizable.

---

## Editor Comment (EC1) · T. Bolch (Editor) · 2 Aug 2016

Dear authors,

I have carefully read the reviewers comments and your manuscript. Both reviewers highlight that the study could potentially of interest as no study of the glacier changes exists for the investigated region but both reviewers also identify several major short-comings and recommend to reject the manuscript for The Cryosphere.

The major identified shortcomings are the lack of innovation and the small spatial cov-erage of the study region. Both issues were also critically mentioned by my access

review. Further identified major issues are the suitability of the selected scenes and the insufficient consideration of the glacier response time.

I am in general in line with the reviewers and feel sorry to reject the manuscript at this stage. This is also to give you enough time to consider the comments. You may nevertheless provide a reply to the reviewers comments.

I encourage you to revise your study and manuscript and I am willing to act as an editor again in case you decide to resubmit a carefully revised manuscript to The Cryosphere. Most important to be accepted for The Cryosphere are to extend the spatial coverage (e.g. whole Tekes River basin or whole Kazakh Tien Shan glaciers) and include more periods (e.g. including data from ∼2000 and/or KH4A data). The climatological analysis should either be deemphasised or the glacier response time need to be better considered. While I do not agree with all comments of the reviewers (e.g. there are only minor language issues and "buffer method" might not be the best, but is a widely accepted method for an estimate of the uncertainty), both reviewers provide suggestions for improvements and detailed comments which should be quite useful for the revision of the manuscript.

Let me know in case you have any questions.

Best regards,

Tobias Bolch – Editor TC

---

## Author Comment (AC1) · 12 Aug 2016

We are grateful to the Reviewers' and Editor's comments and particularly to Dr. Paul's suggestions which we find very helpful. We agree with two main points that the number of glaciers is small and that additional imagery can be used. We will expand the study to the whole of the Tekes basin and use Landsat 1989 and KH4 for the whole of the Tekes basin to improve the coverage.

Comments by Dr. Paul. The selection of the relatively small area for mapping is due to the fact that there is no information on this specific region while work has been done

on the neighbouring areas. It also provides input data to modelling discharge in rivers Bayankol and Tekes (within the borders of Kazakhstan before it flows into China) and this is an important practical consideration behind this work. We used Landsat 1992 imagery and we accept that the 1989 image is of better quality. Regrettably, it was published quite late into the study and not used for this reason (except an unfortunate mistake in Fig. 3). A comparison of glacier area derived from both images, however, shows that the difference is small – 1.6% - and this is well within the uncertainty of measurements. The fact that most glaciers have northern aspect does not disqualify this study. Various ranges in the Tien Shan have glaciers with predominant aspect, e.g. northern aspect predominates in the ZailiiskiyAlatay to which comparisons are made and in this case, it's glacier size that is a controlling factor. However, we accept that maybe a more detailed analysis of the published literature and comparisons with other regions based on aspect are required. We reject the comments on uncertainty assessment. The uncertainty of measurements results from the uncertainty of mapping by individual operator and geolocation and the buffer methods modified by Bolch et al. (2010) from Granshaw and Fountain (2006) is an accepted way of quantifying the latter. While geolocation may be ignored when glacier areas are mapped using imagery for a single year, neglecting it when measuring glacier change using different images from different years, sensors, etc is plain wrong as there will always be an uncertainty term, resulting from mis-registration of two images because of their different geometry, etc. Even when working with ASTER form 2000 and 2010 where there was no change in sensor, assessment of uncertainty on co-registration was required because different DEMs were used in 2000 and 2010 (Shahgedanova et al., 2014). So the correct way is to calculate the difference between the scalar area values as Dr. Paul suggested and add an uncertainty term on image co-registration calculated for both images although this will inevitably lead to higher uncertainty.

Comments by Reviewer 1 We thank Reviewer 1 for the suggestion to use KH4 imagery.Unfortunately, this imagery was not freely available at the time of data processing. It is of better quality than KH9 and will be used for mapping of the whole Tekes

basin. We reject comments by Reviewer 1 on the style of the paper as their command of English language does not qualify them to make such assessments. We point out that the style is offensive and this is both undeserved and unacceptable.

References Bolch, T., Menounos, B. and Wheate, R.: Landsat-based inventory of glaciers in wesetrn Canada, 1985-2005, Remote Sens. Environ., doi:10.1016/j.rse.2009.08.015, 2010. Granshaw, F. D. and Fountain, A. G.: Glacier change (1958-1998) in the North Cascades National Park Complex, Washington, USA, J. Glaciol., 52(177), 251–256, doi:10.3189/172756506781828782, 2006. Shahgedanova, M., G. Nosenko, S.Kutuzov, O.Rototaeva, and T. Khromova:Deglaciation of the Caucasus Mountains, Russia/Georgia, in the 21st century observed with ASTER satellite imagery and aerial photography.The Cryosphere, 8, 2367-2379, 2014.